# EFFORTLESS EVENT-AUGMENTED LATENT DIFFUSION FOR VIDEO FRAME INTERPOLATION

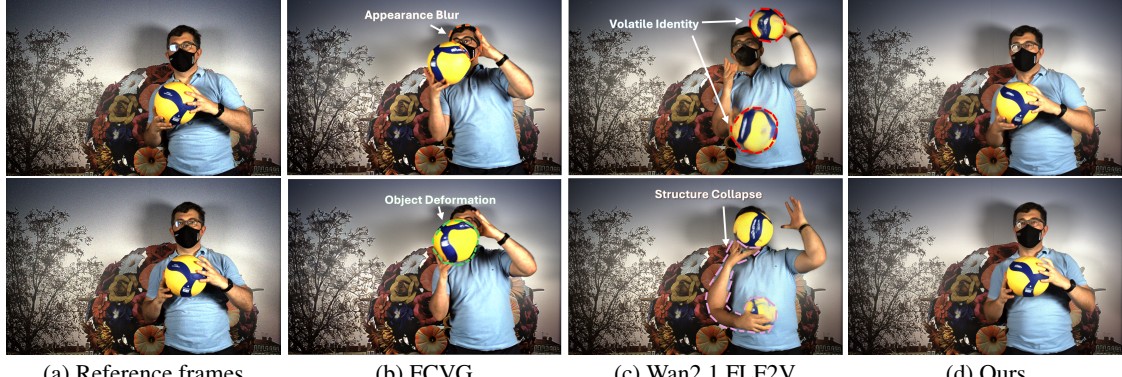

(a) Reference frames      (b) FCVG      (c) Wan2.1 FLF2V      (d) Ours

Figure 1: We propose an effortless event-augmented approach for Diffusion Transformer (DiT)-based video frame interpolation, seamlessly integrating event streams to generate clear and temporally consistent frames. (a) shows the ground-truth intermediate frames at different time stamps for reference. Compared to FCVG Zhu et al. (2024) and Wan2.1 FLF2V Wan et al. (2025), which rely solely on the start and the end frames and often result in motion blur, object deformation, identity inconsistency, and structural artifacts. In contrast, our approach produces sharper and more natural results.

## ABSTRACT

Latent Diffusion Models have advanced video frame interpolation by generating intermediate frames between input frames. However, effectively handling large temporal gaps and complex motion remains a challenge, often leading to artifacts. We argue that event camera signals, with their ability to capture continuous motion at high temporal resolutions, are ideal for bridging these temporal gaps and enhancing interpolation precision. Given the impracticality of training an event-assisted model from scratch, we introduce a novel adapter-based framework that seamlessly and effortlessly integrates high-temporal-resolution cues from event cameras into pre-trained image-to-video models without modifying their underlying structure. Our method leverages Image Warped Events (IWEs) and bidirectional sparse optical flow for precise spatial and temporal alignment, significantly reducing artifacts and improving interpolation quality. Experimental results demonstrate that our event-enhanced interpolation achieves superior accuracy and temporal coherence compared to existing state-of-the-art methods.

## 1 INTRODUCTION

Latent Diffusion Models (LDMs) have recently made significant strides in both image and video generation, spurring advances in video frame interpolation, which involves synthesizing intermediate frames between

start and end frames. Leveraging pre-trained image-to-video (I2V) diffusion models, recent methods can address challenges such as complex motion and large temporal gaps—scenarios where traditional interpolation techniques, which rely on motion estimation and motion compensation, often falter.

For example, recent methods such as GI Wang et al. (2024) and FCVG Zhu et al. (2024) leverage the generative capabilities of Stable Video Diffusion (SVD) Blattmann et al. (2023), which is based on a U-Net diffusion architecture. More recently, Wan2.1 FLF2V Wan et al. (2025) utilizes a powerful Diffusion Transformer (DiT)-based video diffusion model, achieving superior performance in interpolation across substantial temporal gaps. However, despite these advances, LDM-based interpolation methods still suffer from noticeable artifacts, particularly in the intermediate frames of the generated sequences, as illustrated in Fig. 1. We attribute these limitations to the reliance solely on start and end frames for guidance, constraining the interpolation quality.

Event cameras, which asynchronously capture pixel-wise brightness changes, offer unique advantages, including high temporal resolution, broad dynamic range, and low latency. Unlike traditional cameras with fixed frame rates, event cameras provide continuous, high-temporal-resolution motion information that can enrich frame interpolation, especially in complex, high-speed scenes Tulyakov et al. (2021; 2022); Sun et al. (2023); Liu et al. (2024).

However, integrating raw event streams into the I2V diffusion models is non-trivial because event data is sparse, asynchronous, and lacks direct compatibility with the dense, grid-based representations used in mainstream generative models. Furthermore, large-scale paired event-video datasets are scarce, making end-to-end supervised training infeasible.

To address these challenges, we propose to extract Image Warped Events (IWEs) and bidirectional sparse optical flow from event streams using contrast maximization techniques Stoffregen & Kleeman (2019); Shiba et al. (2024). These representations translate the event modality into edge-like and motion-consistent cues that closely align with control signals commonly used in diffusion-based video generation, such as edge maps, flow fields Karmokar et al. (2025); Jiang et al. (2025); Burgert et al. (2025). This serves as a conceptual and practical bridge between event-based vision and frame-based video diffusion.

Building on this insight, we propose a novel adapter-based framework that seamlessly injects motion-aware signals derived from events into a pre-trained video diffusion model. Our method requires only minimal fine-tuning on limited event-video data and does not alter the underlying diffusion architecture. Specifically, we introduce two plug-in adapters: (1) An IWE encoder, which embeds edge-consistent spatial structure into the input latent space; (2) A flow-based alignment-and-fusion adapter, which warps latent features using bidirectional flow before the DiT block and fuses them to form temporally aligned intermediate representations. These adapters inject event-derived structural and temporal cues into the generative process, enhancing interpolation quality and reducing artifacts. To facilitate broader generalization and benchmarking, we also construct a large-scale synthetic event-video dataset, EvPexels, comprising 1,100 diverse scenes (about 390,000 RGB frames) spanning a wide range of motions. To the best of our knowledge, EvPexels is the largest synthetic dataset specifically designed for event-based video frame interpolation. We will release the dataset along with accompanying tools to support future research in event-driven video generation and diffusion modeling.

In summary, our contributions are:

1. We propose a plug-and-play adapter framework that incorporates event-derived signals into DiT-based video diffusion models for frame interpolation, improving temporal consistency and reducing artifacts.

2. We bridge the gap between event streams and LDM-compatible control signals by extracting IWEs and bidirectional optical flow, enabling seamless integration into mainstream generative pipelines.

3. We construct a synthetic event-video dataset with 1,100 diverse motion-rich scenes to support training of event-aware frame interpolation models.

4. Extensive experiments validate the effectiveness of our method, showing superior interpolation quality compared to state-of-the-art baselines.

## 2 RELATED WORK

This section provides an overview of research efforts closely related to our work. We begin by reviewing traditional video frame interpolation techniques, including both frame-based and event-guided approaches. We then examine recent developments in the emerging generative paradigm, with a focus on diffusion-based interpolation methods.

### 2.1 TRADITIONAL VIDEO FRAME INTERPOLATION

Video Frame Interpolation (VFI) is a technique used to reconstruct intermediate frames from a pair of input frames Huang et al. (2022); Kong et al. (2022); Li et al. (2023b); Zhang et al. (2023); Niklaus et al. (2017); Bao et al. (2019). While traditional VFI methods perform well in scenarios with simple motion, they often struggle with complex motions or substantial scene changes between frames.

Event streams, which capture fine-grained motion details between frames, provide more accurate motion estimation for VFI, making event-based VFI methods increasingly popular Tulyakov et al. (2021; 2022); Yu et al. (2021); He et al. (2022); Zhang & Yu (2022); Kim et al. (2023); Sun et al. (2023); Lin et al. (2023). For example, Time Lens Tulyakov et al. (2021) introduced the first VFI model combining warping and synthesis-based approaches. More recently, CBMNet Kim et al. (2023) and TimeLens-XL Ma et al. (2024) have advanced the state of the art by significantly improving the performance of event-based VFI.

While event-guided VFI has improved motion estimation accuracy, these methods still encounter challenges with significant scene changes, such as the appearance of new objects, where event data alone may be insufficient. Consequently, the performance of traditional VFI methods in real-world scenarios involving complex motion and scenes still requires further refinement, motivating diffusion model-based VFI methods, which we discuss next.

### 2.2 DIFFUSION-BASED VIDEO FRAME INTERPOLATION

Diffusion-based VFI techniques have garnered attention due to their ability to handle large and ambiguous motions between frames more effectively than traditional methods. Early work, such as MCVD Voleti et al. (2022), employed latent diffusion models (LDMs) for video prediction and interpolation. Building on this, LDMVFI Danier et al. (2024) applied LDMs specifically for frame interpolation, while VIDIM Jain et al. (2024) advanced this by training diffusion models on larger datasets to enhance performance. CBBD Lyu et al. (2024) introduced the Consecutive Brownian Bridge Diffusion model, which reduces cumulative variance based on the Brownian Bridge Diffusion Model framework Li et al. (2023a). Similarly, Dream-Mover Shen et al. (2024) utilizes stable diffusion priors to interpolate frames with large motions.

However, most of the above diffusion-based models rely on image-to-image (I2I) diffusion frameworks, requiring complex architecture designs and specialized training on specific video datasets. This approach often overlooks recent advancements in I2V models. The emergence of large-scale I2V diffusion models offers a more efficient alternative: adapting pre-trained models (e.g., Stable Video Diffusion Blattmann et al. (2023), Wan2.1 Wan et al. (2025)) for VFI with minimal modifications, enabling training-free or tuning-free methods. Such approaches capitalize on the potential of pre-trained I2V diffusion models for video generation. For instance, TRF Feng et al. (2024) adapts a video generation model for bounded generation,

using initial and final frames to synthesize intermediate frames. However, TRF Feng et al. (2024) does not fully address motion consistency between frames, prompting recent work, such as GI Wang et al. (2024), to introduce a reverse motion method to improve frame-to-frame coherence. More recently, ViBiD Yang et al. (2024), FCVG Zhu et al. (2024) and Wan2.1 FLF2V Wan et al. (2025) have further advanced performance in this domain. Nonetheless, most prior methods have underappreciated the importance of event signals for modeling fine-grained temporal dynamics. Recent efforts—such as U-Net–based LDM approaches Chen et al. (2024)—begin to address this gap, but our strategy is fundamentally different. Rather than training directly on raw event streams, we convert the spatial and motion cues encoded by events into a representation that integrates seamlessly with a DiT-based interpolation model, enabling effective event guidance without requiring end-to-end event-centric training.

## 3 METHODOLOGY

In this section, we first provide an overview of event-based video frame interpolation. We then introduce our fine-tuning pipeline, which involves extracting motion information—specifically, IWEs and bidirectional optical flow—from raw event streams. Next, we present the design of the IWE encoder, which injects edge-aligned spatial features into the video diffusion model, and the alignment and fusion adapter, which utilizes the bidirectional optical flow to warp latent features.

### 3.1 PRELIMINARIES

#### 3.1.1 I2V MODELS FOR VIDEO FRAME INTERPOLATION

The I2V model based on latent diffusion primarily relies on the start frame $I_0$ and the end frame $I_1$ to perform video interpolation. Let the input video sequence be denoted as

$$\mathbf{I} = \{I_0, I_{1/N}, \ldots, I_{(N-1)/N}, I_1\}, \tag{1}$$

where $i \in [0, 1]$ represents normalized time steps. Following Wan2.1 Wan et al. (2025), we adopt a video VAE that compresses the temporal resolution by a factor of 4. Accordingly, the sequence is encoded into a latent video representation

$$\mathbf{Z} = \{z^0, z^1, \ldots, z^k, \ldots, z^{T-1}\}, \tag{2}$$

where $T = \lfloor (N+1)/4 \rfloor + 1$, and $k \in [0, T-1]$.

To avoid ambiguity, all frame indices refer to the latent space throughout the remainder of this paper, unless otherwise specified. The base video diffusion model we use, FLF2V in Wan2.1 Wan et al. (2025), is trained to predict a constant velocity vector $v_t$ from a noisy latent representation $x_t^k$, the timestamp $t$, and the corresponding text condition $c_{\text{txt}}$. The training objective is formulated as the mean squared error (MSE) between the predicted velocity $u(x_t^k, c_{\text{txt}}, t; \theta)$ and the ground-truth $v_t$:

$$\mathcal{L} = \mathbb{E}_{k,t,c_{\text{txt}}} \left\| u(x_t^k, c_{\text{txt}}, t; \theta) - v_t \right\|^2, \tag{3}$$

where $\theta$ denotes the model parameters. This objective guides the model to learn continuous trajectories in latent space, conditioned on the input prompt.

#### 3.1.2 EVENT-ASSISTED VIDEO FRAME INTERPOLATION

To enhance video frame interpolation, we incorporate an event stream

$$\mathcal{E} = \{e_i = (x_i, y_i, \tau_i, p_i)\} \tag{4}$$

between frames $I_0$ and $I_1$. Each event $e_i \in \mathcal{E}$ occurs at spatial coordinates $(x_i, y_i)$, at time $\tau_i$, with polarity $p_i \in \{-1, +1\}$, capturing sparse spatiotemporal changes in the scene. Accordingly, the training objective for the event-assisted interpolation task is modified to:

$$\mathcal{L} = \mathbb{E}_{k,t,\mathcal{E},c_{\text{txt}}} \left\| u(x_t^k, \mathcal{E}, c_{\text{txt}}, t; \theta) - v_t \right\|^2. \tag{5}$$

We first apply contrast maximization (CMax) to compute bidirectional optical flows $\mathbf{f}_{k-1 \to k}$ and $\mathbf{f}_{k+1 \to k}$ from the event intervals $[k-1, k]$ and $[k+1, k]$, respectively:

$$\begin{aligned}
\mathbf{f}_{k-1 \to k}, \ \mathcal{W}^{k-1 \to k} &= \text{CMax}(\mathcal{E}_{[k-1,k]}), \\
\mathbf{f}_{k+1 \to k}, \ \mathcal{W}^{k+1 \to k} &= \text{CMax}(\mathcal{E}_{[k+1,k]}),
\end{aligned} \tag{6}$$

where $\mathcal{E}_{[k-1,k]}$ and $\mathcal{E}_{[k+1,k]}$ denote the subsets of events occurring between the respective frames. The outputs $\mathcal{W}^{k-1 \to k}$ and $\mathcal{W}^{k+1 \to k}$ are IWE representations warped by forward and backward optical flow. We employ an IWE encoder to extract the aligned spatial edge features from the IWE maps, denoted as $\mathbf{F}_{\mathcal{W}}^k$, which are subsequently injected into the input latent representation to enhance structural guidance.

In the alignment and fusion module, we warp the DiT features from the adjacent latent states $\mathbf{F}_{x_t}^{k-1}$ and $\mathbf{F}_{x_t}^{k+1}$ toward the current frame $k$, using the estimated optical flows:

$$\begin{aligned}
\mathbf{F}_{x_t}^{k-1 \to k} &= \text{Warp}(\mathbf{F}_{x_t}^{k-1}, \ \mathbf{f}_{k \to k-1}), \\
\mathbf{F}_{x_t}^{k+1 \to k} &= \text{Warp}(\mathbf{F}_{x_t}^{k+1}, \ \mathbf{f}_{k \to k+1}).
\end{aligned} \tag{7}$$

These aligned features are then fused with the center frame's features $\mathbf{F}_{x_t}^k$ via a fusion function $G(\cdot)$:

$$\mathbf{F}_{\text{fused}}^k = G\left(\mathbf{F}_{x_t}^k, \ \mathbf{F}_{x_t}^{k-1 \to k}, \ \mathbf{F}_{x_t}^{k+1 \to k}\right). \tag{8}$$

Finally, the fused residual features of all frames $\mathbf{F}_{\text{fused}}$ are added to the DiT features to enhance temporal consistency across frames.

## 3.2 FRAMEWORK

As shown in Fig. 2, our framework takes the start frame $I_0$, the end frame $I_1$, and the corresponding event stream as input, and outputs the intermediate frames $\hat{\mathbf{I}}$. The overall architecture consists of an event representation module and an adapter-enhanced fine-tuning strategy.

### 3.2.1 EVENT REPRESENTATION

To extract motion cues from the event stream, we follow the principle of contrast maximization Stoffregen & Kleeman (2019); Shiba et al. (2024), which enables both optical flow estimation and generation of the Image Warped Events (IWE)—a sharp, edge-aware image obtained by temporally aligning events at a designated reference time. We employ an off-the-shelf CMax-based method Shiba et al. (2024) to compute optical flows and their corresponding IWEs. For temporal alignment, we divide the event stream between $I_0$ and $I_1$ into multiple temporal segments. For each segment $[k-1, k]$, we compute a sparse forward optical flow $\mathbf{f}_{k-1 \to k}$ and the corresponding IWE $\mathcal{W}^{k-1 \to k}$, by masking the dense flow with event activity. Similarly, we reverse the event stream and extract backward flows $\mathbf{f}_{k+1 \to k}$ and IWEs $\mathcal{W}^{k+1 \to k}$. This process yields bidirectional sparse optical flows and edge-aware IWE representations across the entire sequence.

**IWE-Based Spatial Conditioning** To extract structured spatial cues from the event stream, we design an encoder that processes the bidirectional IWEs and produces aligned feature maps $\mathbf{F}_{\mathcal{W}}^k$, where $k \in [0, T]$. IWEs are known to correlate strongly with scene edges and object boundaries Karmokar et al. (2025), making them particularly effective for guiding frame synthesis in motion-intensive regions. Given the sparse and

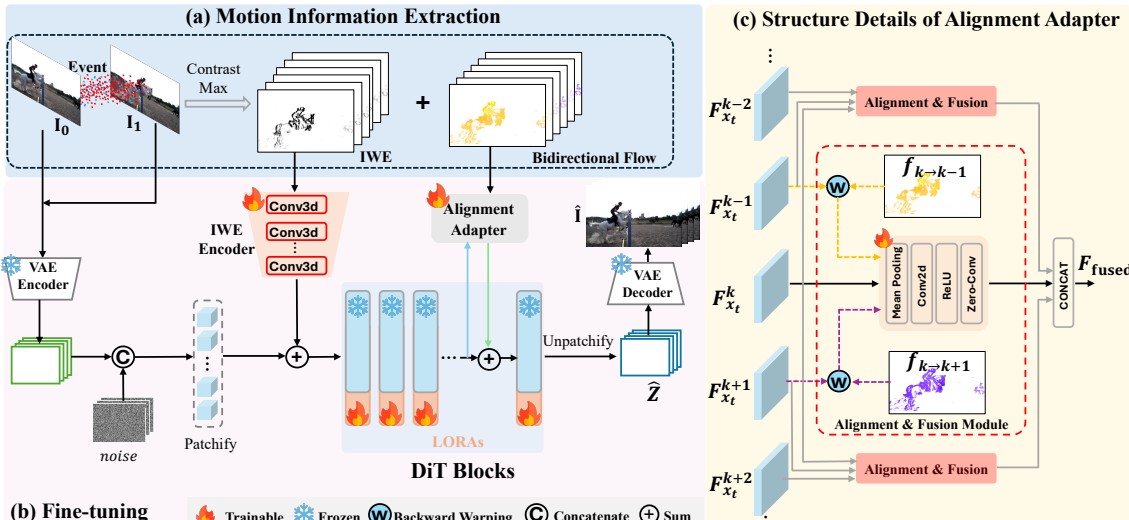

Figure 2: Illustration of Our Framework. (a) We extract bidirectional sparse optical flow and IWEs from the input event stream using the Contrast Maximization (CMax) method. (b) During fine-tuning, the model is enhanced with three components: an IWE encoder, alignment adapters inserted into a subset of DiT blocks, and LoRA layers applied to all DiT blocks. (c) The flow-based alignment adapter leverages the bidirectional flows to warp intermediate features from neighboring frames, aligning them temporally with the current frame. This facilitates motion-consistent feature propagation throughout the denoising process.

lightweight nature of the IWE signal, we adopt a simple yet effective integration strategy. The IWE maps are encoded into feature representations using a compact network composed of 3D convolutional layers. These features are then injected into the video latent space via element-wise addition to the input latents of the diffusion model. This design enables edge-aware spatial conditioning with minimal computational overhead. To adapt the pre-trained diffusion model to this new input modality, we apply LoRA-based fine-tuning across all DiT blocks. This allows the model to leverage IWE-derived structural features while keeping the majority of the original parameters frozen, ensuring parameter efficiency and architectural compatibility.

**Flow-Based Temporal Alignment and Fusion** Before each selected DiT block, we rearrange the patch-based latent representations back into a frame-wise format that is spatially aligned with the optical flows. To enforce temporal consistency, we first reshape the bidirectional optical flows and warp the features from the previous and next frames toward the current frame using the corresponding forward and backward flows. The three aligned features (from frames $k-1$, $k$, and $k+1$) are then aggregated via mean pooling and passed through a lightweight fusion network composed of convolutional layers. The fused feature is subsequently added to the original DiT feature as a residual correction, enhancing both spatial fidelity and temporal coherence. Finally, the updated frame-wise features are rearranged back into patch tokens and propagated to the selected DiT block for further refinement. This design allows the model to benefit from explicit motion guidance without incurring significant computational overhead. To balance performance and efficiency, the alignment-and-fusion module is only applied to a subset of DiT blocks rather than all layers.

## 4 EXPERIMENTS

### 4.1 EXPERIMENTAL SETUP

**Training Datasets** We fine-tune our framework using two datasets: the real-world BS-ERGB dataset Tulyakov et al. (2022) and our curated synthetic EvPexels dataset. The BS-ERGB dataset contains high-speed image-event pairs captured at a resolution of 970×625 and a frame rate of 28 fps. The training split includes 48 video clips, while the test split consists of 26 clips and is used for quantitative evaluation.

To augment training diversity, we introduce the EvPexels dataset, constructed from videos collected via the Pexels platform (https://www.pexels.com). We select videos exhibiting diverse motion

Table 1: Quantitative comparison of the VFI performance on the BS-ERGB test dataset. **Bold** indicates the best performance under the 24× interpolation setting.

| Methods | BS-ERGB | | | | |
| | PSNR↑ | SSIM↑ | LPIPS↓ | FID↓ | FVD↓ |
|---|---|---|---|---|---|
| TRF Feng et al. (2024) | 14.078 | 0.4117 | 0.426 | 47.146 | 971.424 |
| GI Wang et al. (2024) | 16.964 | 0.518 | 0.311 | 33.082 | 588.371 |
| ViBiD Yang et al. (2024) | 15.525 | 0.475 | 0.352 | 39.027 | 788.652 |
| FCVG Zhu et al. (2024) | 17.809 | 0.546 | 0.302 | 26.832 | 726.752 |
| Wan2.1-FLF2V Wan et al. (2025) | 18.698 | 0.618 | 0.212 | 18.607 | 376.828 |
| TimeLens Tulyakov et al. (2021) | 24.704 | 0.699 | 0.165 | 43.808 | 851.523 |
| CBMNet-Large Kim et al. (2023) | **25.306** | **0.712** | 0.169 | 17.658 | 228.753 |
| TimeLens-XL Ma et al. (2024) | 21.737 | 0.678 | 0.248 | 47.155 | 710.688 |
| Ours | 23.261 | 0.704 | **0.132** | **8.168** | **117.368** |

patterns using TransNet V2 Soucek & Lokoc (2024), ensuring single-shot segments suitable for the frame interpolation task. Event streams are synthesized from RGB videos using the Vid2e simulator Gehrig et al. (2020). The resulting dataset comprises 1,100 video sequences, totaling 389,761 frames at a resolution of 704×480. The visualization of the EvPexels dataset are provided in the Appendix A.1.2.

**Test Datasets** We evaluate performance on a real-captured dataset (BS-ERGB test set) and two synthetic datasets. As for the two additional synthetic datasets, following prior works such as TRF Feng et al. (2024) and GI Wang et al. (2024), we select 50 video clips from the DAVIS dataset Pont-Tuset et al. (2017) and 30 clips from Pexels, each consisting of 25 frames. These datasets cover diverse motion scenarios and provide a comprehensive evaluation of interpolation quality.

**Implementation Detail** We adopt the open-source FLF2V model from Wan2.1 Wan et al. (2025) as our base video diffusion architecture. The model takes the first and last frames as input and generates a video of fixed length (81 frames) in latent space. The learning rate is set to $1 \times 10^{-4}$, and all input images are resized to a resolution of 832×480 during training. Other hyperparameters follow the original FLF2V configuration without modification. In our experiments, optical flow information is injected into two DiT blocks to enhance motion guidance. We fine-tune our model for 4,000 steps on 8 NVIDIA A800 GPUs with a global batch size of 8.

**Evaluation Metrics** We calculate metrics including PSNR (Peak Signal-to-Noise Ratio), SSIM (Structural Similarity Index Measure), Learned Perceptual Image Patch Similarity (LPIPS), Fréchet Inception Distance (FID), and Fréchet Video Distance (FVD).

### 4.2 QUANTITATIVE & QUALITATIVE EVALUATION

**Quantitative Evaluation** To comprehensively evaluate our method, we compare it against a broad range of video frame interpolation (VFI) baselines. For event-based VFI, we include TimeLens Tulyakov et al. (2021), CBMNet-Large Kim et al. (2023), as well as the more recent TimeLens-XL Ma et al. (2024) and VDM-EVFI Chen et al. (2024). For frame-based VFI, we consider several diffusion-based generative approaches, including TRF Feng et al. (2024), GI Wang et al. (2024), ViBiD Yang et al. (2024), FCVG Zhu et al. (2024), and Wan2.1 FLF2V Wan et al. (2025).

For methods with publicly available training code—namely Wan2.1 FLF2V Wan et al. (2025), VDM-EVFI Chen et al. (2024), CBMNet-Large Kim et al. (2023), and TimeLens-XL Ma et al. (2024)—we fine-

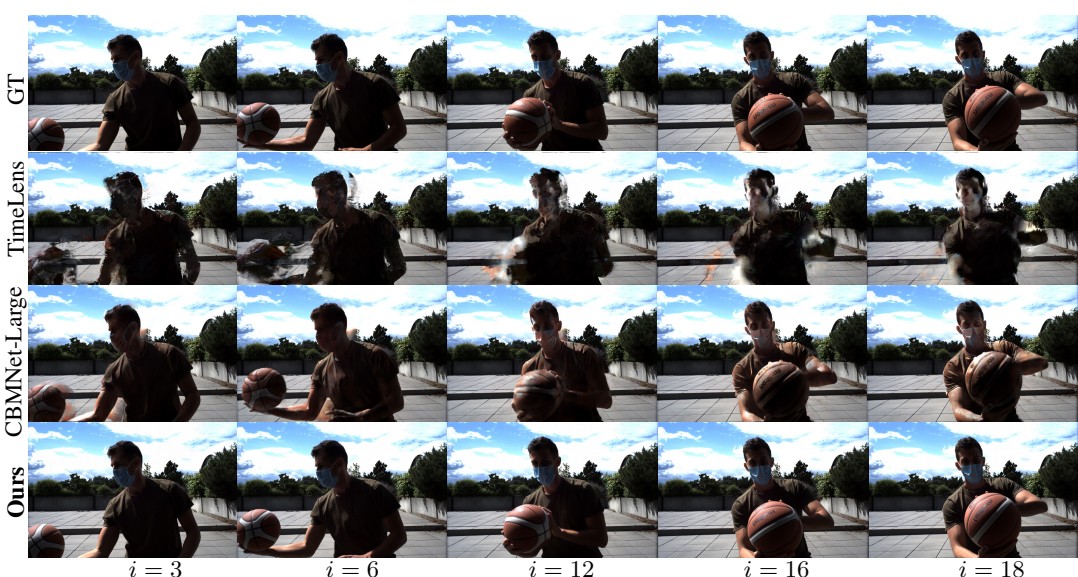

Figure 3: Visual comparison of VFI methods on the BS-ERGB test dataset (time $\times$ 24).

tune the official pretrained models on our training datasets to ensure a fair comparison. For methods without released training code, such as TimeLens Tulyakov et al. (2021), TRF Feng et al. (2024), GI Wang et al. (2024), ViBiD Yang et al. (2024), and FCVG Zhu et al. (2024), we use their official checkpoints for inference. Following standard practice in diffusion-based interpolation Feng et al. (2024); Wang et al. (2024), all methods (except VDM-EVFI Chen et al. (2024)) are evaluated under a $\times 24$ interpolation setting across three datasets: BS-ERGB, DAVIS, and Pexels. The quantitative results on the BS-ERGB test set with $\times 24$ interpolation are summarized in Tab. 1, while the results on DAVIS and Pexels are reported in Tab. 2. Since VDM-EVFI supports only $\times 12$ interpolation, we additionally compare against it under this setting, with results shown in Tab. 3.

In the $\times 24$ interpolation setting, our method achieves the best overall performance across all metrics—PSNR, SSIM, LPIPS, FID, and FVD—on the DAVIS and Pexels datasets. On BS-ERGB test dataset, our approach achieves state-of-the-art results on perceptual metrics (LPIPS, FID, and FVD), while ranking third and second in PSNR and SSIM, respectively, among distortion-based metrics. Traditional event-based methods, such as TimeLens Tulyakov et al. (2021) and CBMNet-Large Kim et al. (2023), achieve higher PSNR scores, with CBMNet-Large also showing strong SSIM performance on BS-ERGB. This is largely due to the design of conventional interpolation methods, which emphasize pixel-level reconstruction accuracy. Consequently, they perform well on distortion-based metrics (e.g., PSNR, SSIM), but often fall short in generating perceptually realistic or temporally consistent frames. By contrast, generative models prioritize visual realism and temporal consistency, leading to superior perceptual quality even if pixel-wise similarity is sometimes compromised. The qualitative comparisons further illustrate these trends.

In the $\times 12$ interpolation setting, our method consistently outperforms VDM-EVFI Chen et al. (2024) across nearly all evaluation metrics, particularly those based on perception. While our approach achieves a slightly lower PSNR (0.16 dB less) on BS-ERGB, it demonstrates significantly stronger generalization ability, as evidenced by superior results on the DAVIS and Pexels datasets. Additional visual comparisons in the Appendix A.2.3 further validate the effectiveness of our method.

Table 2: Quantitative comparison on the VFI tasks on DAVIS and Pexels datasets (time $\times$ 24).

| Methods | DAVIS | | | | | Pexels | | | | |
|---|---|---|---|---|---|---|---|---|---|---|
| | PSNR↑ | SSIM↑ | LPIPS↓ | FID↓ | FVD↓ | PSNR↑ | SSIM↑ | LPIPS↓ | FID↓ | FVD↓ |
| TRF Feng et al. (2024) | 14.132 | 0.459 | 0.484 | 70.528 | 1373.954 | 16.737 | 0.600 | 0.400 | 109.516 | 1624.791 |
| GI Wang et al. (2024) | 14.850 | 0.467 | 0.406 | 55.067 | 1158.330 | 17.700 | 0.600 | 0.306 | 109.029 | 1212.097 |
| ViBiD Yang et al. (2024) | 14.811 | 0.456 | 0.448 | 55.343 | 1194.670 | 17.413 | 0.588 | 0.365 | 104.089 | 1335.211 |
| FCVG Zhu et al. (2024) | 16.162 | 0.509 | 0.385 | 48.839 | 1246.823 | 19.172 | 0.635 | 0.275 | 105.617 | 1481.806 |
| Wan2.1-FLF2V Wan et al. (2025) | 17.510 | 0.538 | 0.310 | 36.740 | 800.613 | 19.747 | 0.642 | 0.223 | 46.009 | 959.256 |
| TimeLens Tulyakov et al. (2021) | 22.913 | 0.632 | 0.352 | 102.191 | 1706.523 | 27.071 | 0.757 | 0.215 | 79.886 | 1093.200 |
| CBMNet-Large Kim et al. (2023) | 20.633 | 0.742 | 0.343 | 79.459 | 1164.145 | 23.429 | 0.799 | 0.292 | 81.449 | 840.820 |
| TimeLens-XL Ma et al. (2024) | 17.498 | 0.530 | 0.235 | 100.506 | 1438.467 | 26.241 | 0.789 | 0.224 | 82.760 | 557.176 |
| Ours | **25.544** | **0.799** | **0.115** | **13.367** | **158.557** | **29.089** | **0.858** | **0.080** | **16.319** | **151.345** |

**Qualitative Results**  Fig. 3 shows qualitative comparisons on the BS-ERGB test set, which features challenging motion involving a person and a basketball. Although TimeLens Tulyakov et al. (2021) and CBMNet-Large Kim et al. (2023) attain higher PSNR scores, their visual quality is clearly inferior. TimeLens Tulyakov et al. (2021) suffers from noticeable artifacts near moving objects, while CBMNet-Large Kim et al. (2023) generates a distorted appearance of the basketball. In contrast, our method effectively leverages intermediate event information to accurately model the motion of dynamic foreground objects, resulting in temporally coherent and visually faithful reconstructions. Additional visual results and ablation study are provided in the appendix, further demonstrating the effectiveness of our method.

## 5 CONCLUSION

In this paper, we explore leveraging event data to efficiently enhance DiT-based video frame interpolation tasks. We propose an adapter-based framework that integrates high temporal resolution cues from event cameras—capturing continuous motion data via a pre-trained Image-to-Video (I2V) model, requiring only lightweight adapter training. By incorporating IWE and bidirectional sparse optical flow, our approach enables precise temporal guidance, mitigating motion artifacts and improving interpolation quality. Our experimental results demonstrate that event-enhanced interpolation outperforms existing methods in terms of both accuracy and temporal consistency, effectively reducing long-range motion drift and improving structural fi-

Table 3: Quantitative comparison on the VFI tasks on BS-ERGB, DAVIS and Pexels dataset (time $\times$ 12).

| BS-ERGB | PSNR↑ | SSIM↑ | LPIPS↓ | FID↓ | FVD↓ |
|---|---|---|---|---|---|
| VDM-EVFI | **23.590** | **0.741** | 0.184 | 30.335 | 299.201 |
| Ours | 23.437 | 0.706 | **0.131** | **10.532** | **146.095** |
| **DAVIS** | PSNR↑ | SSIM↑ | LPIPS↓ | FID↓ | FVD↓ |
| VDM-VFI | 23.831 | 0.767 | 0.253 | 52.178 | 510.708 |
| Ours | **25.918** | **0.803** | **0.113** | **17.138** | **183.573** |
| **Pexels** | PSNR↑ | SSIM↑ | LPIPS↓ | FID↓ | FVD↓ |
| VDM-VFI | 27.715 | 0.844 | 0.194 | 42.654 | 401.272 |
| Ours | **29.411** | **0.861** | **0.079** | **18.905** | **168.214** |

delity. This confirms the feasibility of extracting optical flow and IWE from event data to assist frame interpolation, thereby circumventing the challenges associated with directly adapting sparse event data to dense RGB frames.

**Discussion and Future Work:** While our work primarily focuses on utilizing the high temporal resolution of event data to aid frame interpolation, event cameras also possess unique spatial advantages, such as a high dynamic range and robustness to lighting variations. These properties could be further leveraged to enhance video generation under challenging conditions.

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

## A APPENDIX

### A.1 ADDITIONAL EXPERIMENTS

#### A.1.1 ABLATION STUDY

**Impact of Different Input Features** To systematically evaluate the contribution of each input component to the overall interpolation performance, we conduct an ablation study by selectively removing or modifying specific modules in our framework and retraining each variant from scratch. The ablation settings include:

1. **No IWE and flows:** Only fine-tunes the Wan2.1 FLF2V model on our dataset, without incorporating any event-based features (including IWE and flows).

2. **w/o optical flow:** Removes the optical flow warping module while retaining the IWE encoder.

3. **w/o IWE:** Removes the IWE encoder while keeping the optical flow warping and fusion module.

4. **IWE and flow as input:** Uses both IWE and flow features as direct inputs, but disables the warping mechanism.

5. **Full model**: Use IWE as input, and flows as warping manner to align the temporal features.

All ablation variants are individually trained for 5,400 steps on a NVIDIA A800 GPU and evaluated on the Pexels test set. The quantitative results are presented in Tab. 4. The results confirm that both IWE and optical flow features play essential roles in enhancing interpolation quality. Directly injecting flow information as input (without warping) yields inferior performance compared to the warping-based approach, highlighting the effectiveness of explicit temporal alignment via flow-guided feature warping.

Table 4: Ablation study on Pexels test dataset of our method (time $\times$ 24).

| Methods | PSNR↑ | SSIM↑ | LPIPS↓ | FID↓ | FVD↓ |
|---|---|---|---|---|---|
| w/o (IWE and flows) | 16.928 | 0.577 | 0.301 | 59.362 | 1823.590 |
| w/o flows warping | 25.130 | 0.783 | 0.110 | 27.207 | 252.806 |
| w/o IWE | 25.231 | 0.785 | 0.111 | 27.249 | 242.496 |
| IWE & flow inputs | 25.179 | 0.790 | 0.112 | 27.519 | 234.407 |
| Full model | **25.693** | **0.804** | **0.100** | **23.445** | **217.054** |

#### A.1.2 EVPEXELS DATASET

The EvPexels dataset we collected covers a diverse range of motion scenarios, including urban street scenes, natural environments, aerial views, traffic, humans, and pets. Fig. 4 presents visualizations of the corresponding event data and video frames from a subset of these scenarios. Each video, with a spatial resolution of 704×480, consists of a single continuous shot containing no more than 500 frames.

### A.2 ADDITIONAL ANALYSIS

#### A.2.1 VISUALIZATION RESULTS ON SCENES WITH CAMERA MOTION

Additional qualitative results on the DAVIS dataset are presented in Fig. 5, featuring scenes with large camera motion. Our method consistently produces artifact-free, temporally smooth frames, while competing methods exhibit significant visual degradations.

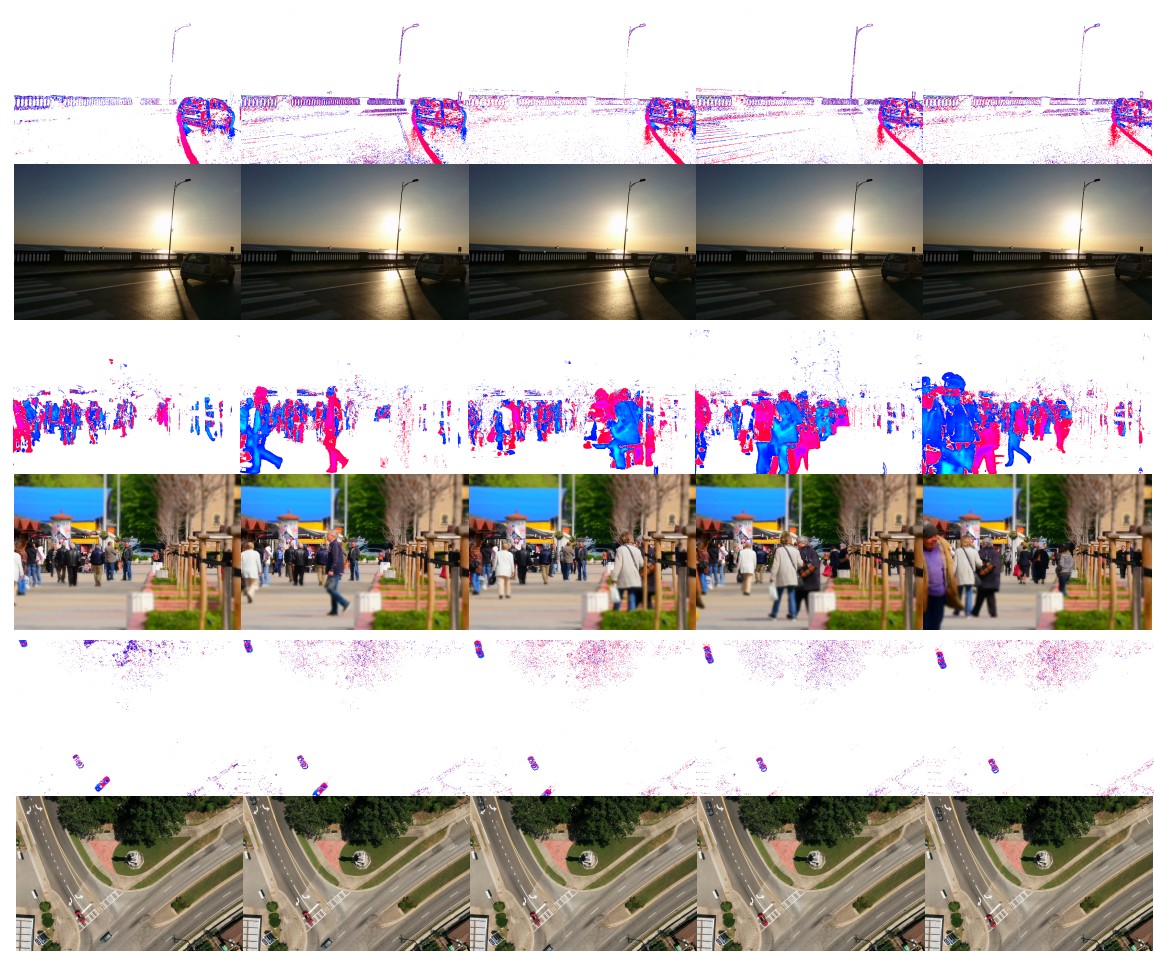

Figure 4: Visualization of the EvPexels dataset.

### A.2.2 QUALITATIVE COMPARISON WITH WAN2.1 FLF2V

Wan2.1 FLF2V Wan et al. (2025), as an advanced DiT-based VFI method, demonstrates promising performance in our experiments. For instance, in the large-motion scenario shown in Fig. 6, even without intermediate motion guidance, Wan2.1 FLF2V is able to generate relatively smooth and visually plausible interpolated frames. However, compared to our approach, certain limitations remain. Our method can better capture intermediate motion, while Wan2.1 FLF2V, relying solely on the start and end frames, fails to accurately follow the ground-truth motion trajectory of the subject. In the generated intermediate frames, the human pose (e.g., the head) sometimes appears distorted, deviating from natural body structure. In the second example (Fig. 7), the interpolation results of Wan2.1 FLF2V even produce artifacts such as multiple limbs and duplicate balls, whereas our method generates temporally coherent and structurally consistent intermediate frames. These two cases indicate that although Wan2.1 FLF2V, as a baseline VFI model, is capable of producing reasonably smooth videos, it still lacks reliable motion guidance in large-motion sce-

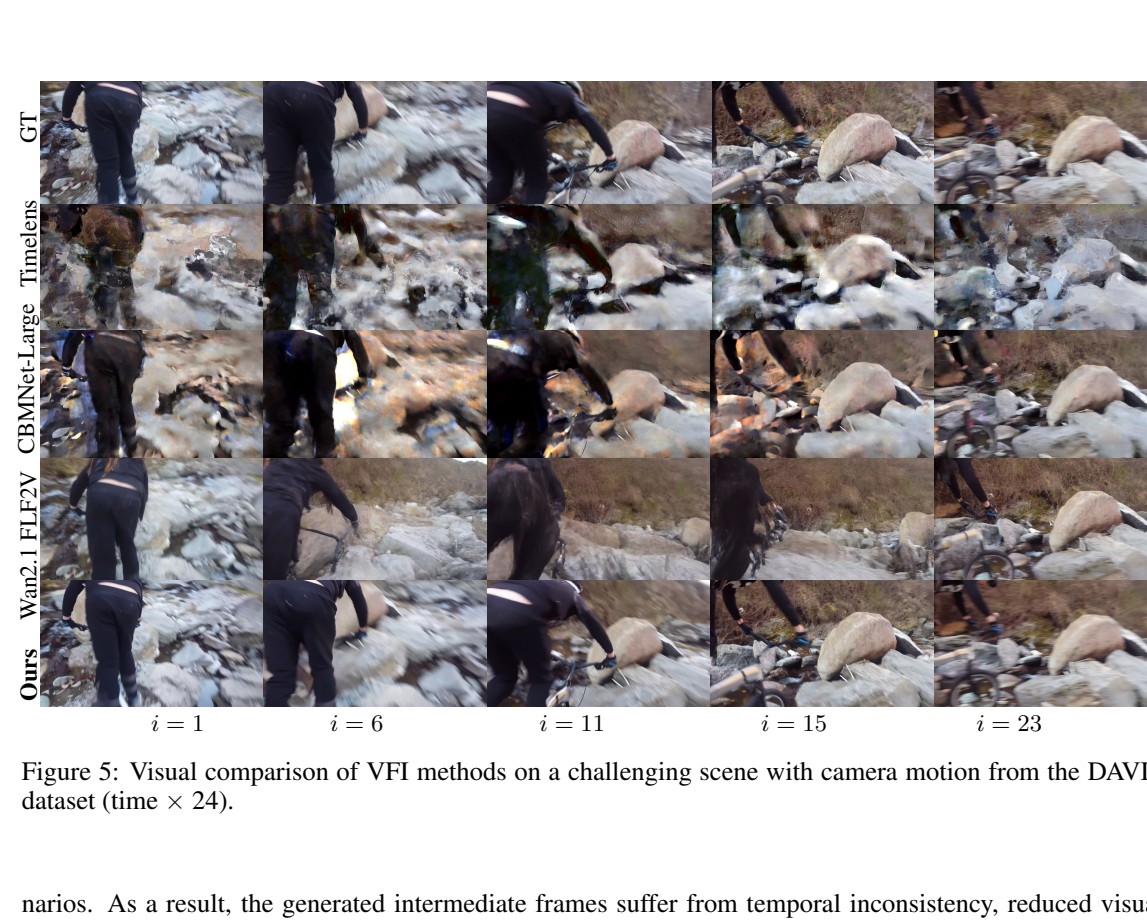

Figure 5: Visual comparison of VFI methods on a challenging scene with camera motion from the DAVIS dataset (time × 24).

narios. As a result, the generated intermediate frames suffer from temporal inconsistency, reduced visual fidelity, and impaired structural preservation.

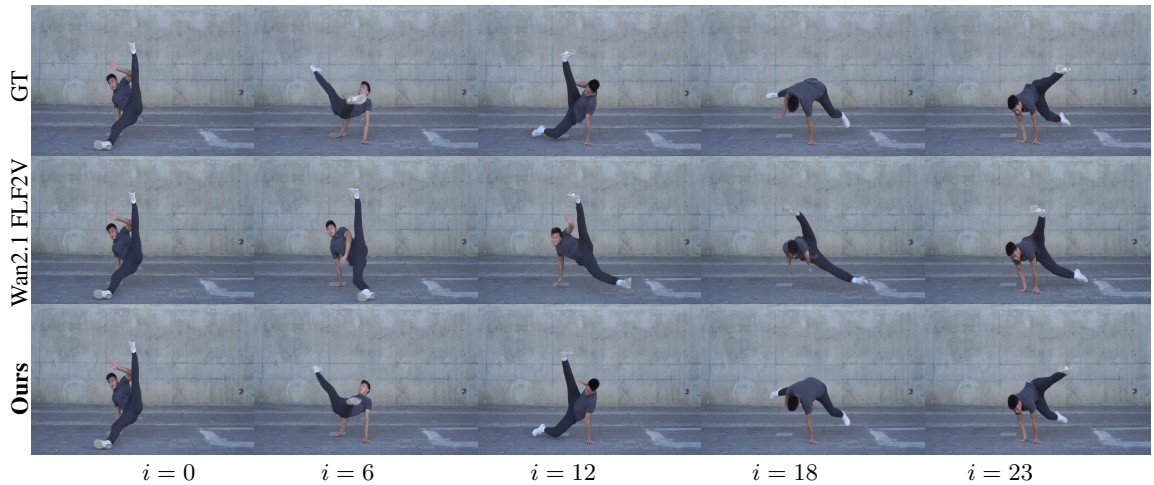

Figure 6: Visual comparison with Wan2.1 FLF2V on the Pexels dataset (time × 24).

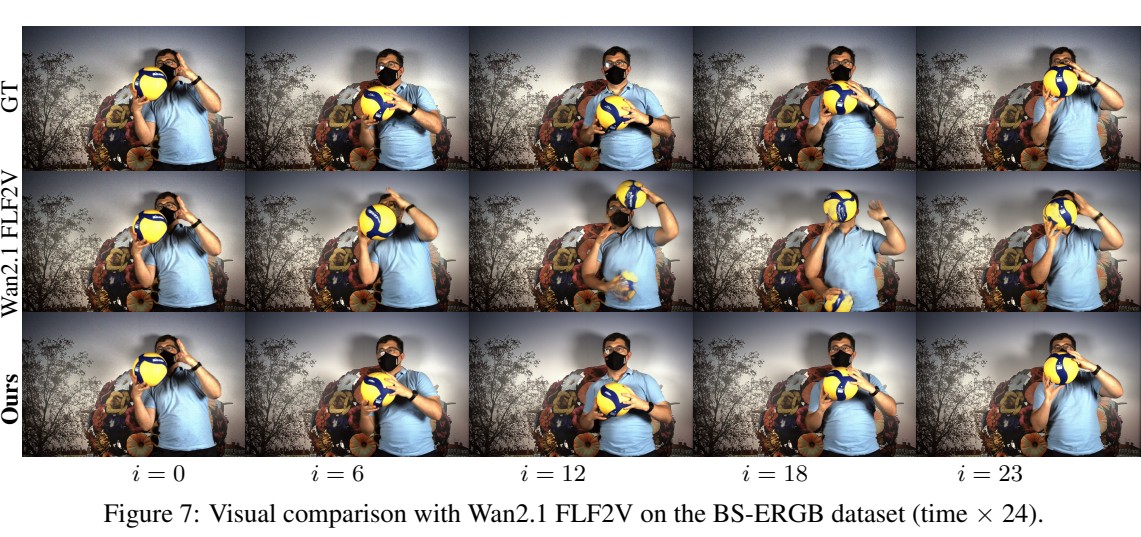

Figure 7: Visual comparison with Wan2.1 FLF2V on the BS-ERGB dataset (time $\times$ 24).

### A.2.3 QUALITATIVE COMPARISON WITH VDM-EVFI

As shown in Tab. 3, VDM-EVFI Chen et al. (2024) achieves slightly higher PSNR and SSIM scores than our method on the BS-ERGB dataset; however, it performs substantially worse on perceptual metrics such as LPIPS, FID, and FVD. The qualitative results in Fig. 8 and 9 further demonstrate this gap. In the $\times$ 12 frame interpolation scenario, our method produces noticeably better results than VDM-EVFI on the BS-ERGB dataset. For instance, in Fig. 8 , VDM-EVFI fails to faithfully reconstruct the subject's facial region and left arm, whereas our method preserves fine details clearly, such as the reflections on the eyeglass lenses. In the large-motion turning scenario shown in Fig. 9, VDM-EVFI struggles to handle facial and clothing details, while the frames generated by our method are visually much closer to the ground truth.

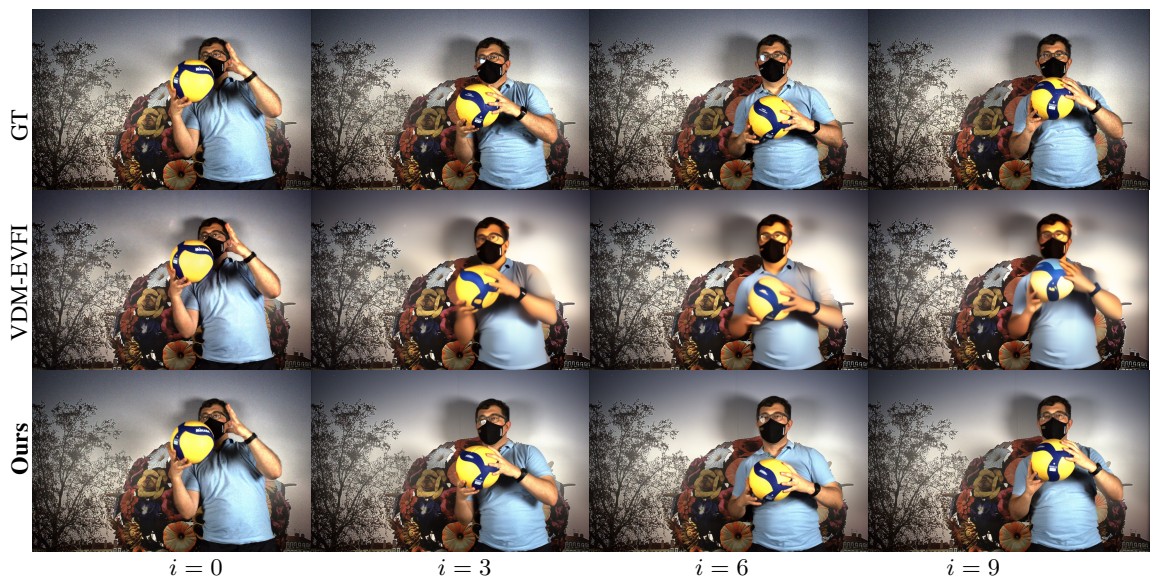

Figure 8: Visual comparison with VDM-EVFI on the BS-ERGB dataset (time $\times$ 12).

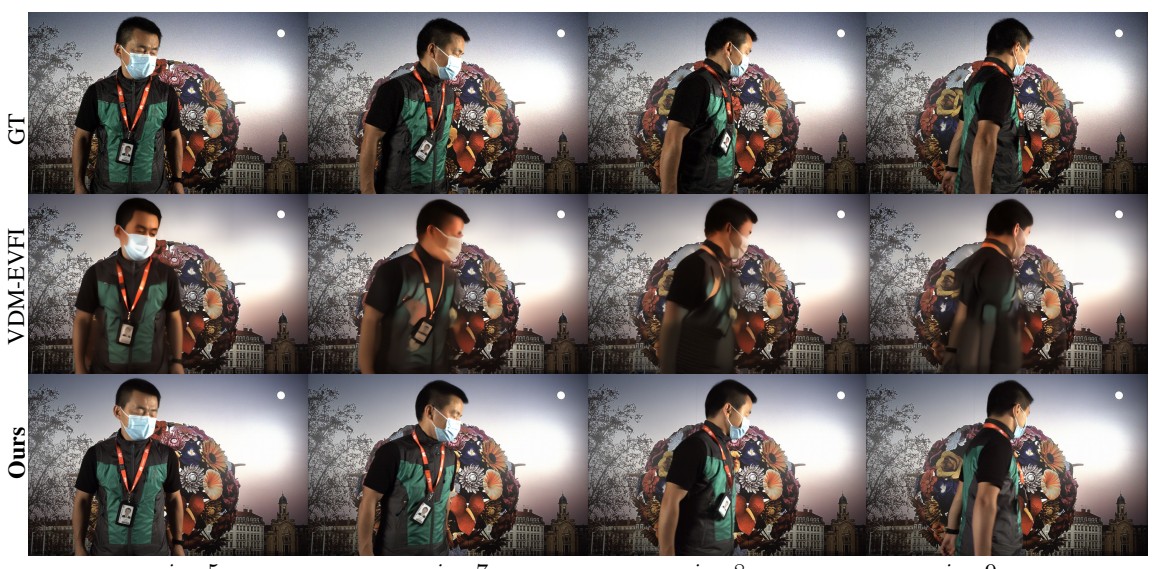

Figure 9: Visual comparison with VDM-EVFI on the BS-ERGB dataset (time $\times$ 12).

### A.2.4 VISUALIZATION OF IWES AND OPTICAL FLOWS

We provide a detailed visualization of the sparse bidirectional optical flow and IWE, as shown in Fig. 10 (b). Specifically, we apply a contrast maximization method on the event data between every two consecutive latent frames to compute these sparse optical flow segments and the IWE. Comparing them with the ground truth in Fig. 10 (a) reveals that the optical flow segments in Fig. 10 (b) accurately capture the motion between consecutive frames in Fig. 10 (a).

### A.2.5 VIDEO RESULTS

Please refer to our project page in the supplementary material: `event_vfi/index.html` for video results, which clearly demonstrate that our reconstructions provide superior consistency and generalization compared to other baselines.

### A.3 THE USE OF LARGE LANGUAGE MODELS(LLMS)

In this study, Large Language Models (LLMs) were employed solely to improve the readability and polish of the manuscript. No part of the substantive analysis, results, or interpretations was generated by LLMs.

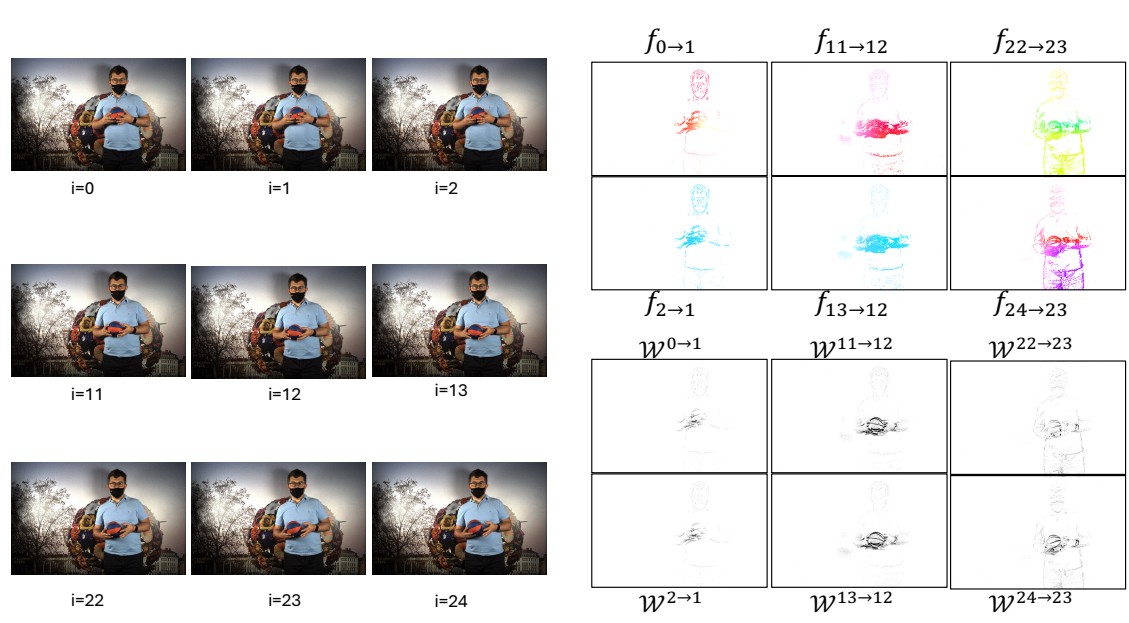

(a) Reference Images

(b) Bidirectional Optical Flow & IWE

Figure 10: Illustration of the optical flow & IWE.

# B REBUTTAL APPENDIX

## B.1 QUANTITATIVE AND QUALITY EVALUATION

For a comprehensive comparison, we adapt the VDM-EVFI Chen et al. (2024) method—originally based on Stable Video Diffusion—to the Wan2.1 FLF2V Wan et al. (2025) backbone, and refer to this variant as VDM-EVFI-Wan2.1. We retrain this model from scratch on our training set. To ensure fairness, we strictly follow the same training configuration as our method, using 8 NVIDIA GPUs and maintaining the same number of training iterations. We further include the non-generative interpolation method RIFE Huang et al. (2022) as an additional baseline. All methods are evaluated under a ×24 interpolation setting across three datasets: BS-ERGB, DAVIS, and Pexels. The quantitative results on the BS-ERGB test set are reported in Tab. 5, while the results on DAVIS and Pexels are presented in Tab. 6. Qualitative visual comparisons are illustrated in Fig. 11 and Fig. 12.

The quantitative and qualitative comparisons show that VDM-EVFI-Wan2.1 remains inferior to our method even when equipped with the same backbone, demonstrating the effectiveness of our design. Furthermore, the non-generative baseline RIFE exhibits clear limitations in large-motion scenarios, which further highlights the advantages of our approach.

Table 5: Quantitative comparison of the VFI performance on the BS-ERGB test dataset. **Bold** indicates the best performance under the 24× interpolation setting.

| Methods | BS-ERGB | | | | |
| --- | --- | --- | --- | --- | --- |
| | PSNR↑ | SSIM↑ | LPIPS↓ | FID↓ | FVD↓ |
| RIFE Huang et al. (2022) | 22.174 | 0.641 | 0.172 | 35.347 | 1113.496 |
| TRF Feng et al. (2024) | 14.078 | 0.4117 | 0.426 | 47.146 | 971.424 |
| GI Wang et al. (2024) | 16.964 | 0.518 | 0.311 | 33.082 | 588.371 |
| ViBiD Yang et al. (2024) | 15.525 | 0.475 | 0.352 | 39.027 | 788.652 |
| FCVG Zhu et al. (2024) | 17.809 | 0.546 | 0.302 | 26.832 | 726.752 |
| Wan2.1-FLF2V Wan et al. (2025) | 18.698 | 0.618 | 0.212 | 18.607 | 376.828 |
| TimeLens Tulyakov et al. (2021) | 24.704 | 0.699 | 0.165 | 43.808 | 851.523 |
| CBMNet-Large Kim et al. (2023) | **25.306** | **0.712** | 0.169 | 17.658 | 228.753 |
| TimeLens-XL Ma et al. (2024) | 21.737 | 0.678 | 0.248 | 47.155 | 710.688 |
| VDM-EVFI-Wan2.1 Chen et al. (2024) | 22.402 | 0.673 | 0.282 | 15.693 | 145.067 |
| Ours | 23.261 | 0.704 | **0.132** | **8.168** | **117.368** |

Table 6: Quantitative comparison on the VFI tasks on DAVIS and Pexels datasets (time × 24).

| Methods | DAVIS | | | | | Pexels | | | | |
| --- | --- | --- | --- | --- | --- | --- | --- | --- | --- | --- |
| | PSNR↑ | SSIM↑ | LPIPS↓ | FID↓ | FVD↓ | PSNR↑ | SSIM↑ | LPIPS↓ | FID↓ | FVD↓ |
| RIFE Huang et al. (2022) | 18.287 | 0.487 | 0.402 | 82.137 | 2028.480 | 21.820 | 0.634 | 0.274 | 80.118 | 1976.327 |
| TRF Feng et al. (2024) | 14.132 | 0.459 | 0.484 | 70.528 | 1373.954 | 16.737 | 0.600 | 0.400 | 109.516 | 1624.791 |
| GI Wang et al. (2024) | 14.850 | 0.467 | 0.406 | 55.067 | 1158.330 | 17.700 | 0.600 | 0.306 | 109.029 | 1212.097 |
| ViBiD Yang et al. (2024) | 14.811 | 0.456 | 0.448 | 55.343 | 1194.670 | 17.413 | 0.588 | 0.365 | 104.089 | 1335.211 |
| FCVG Zhu et al. (2024) | 16.162 | 0.509 | 0.385 | 48.839 | 1246.823 | 19.172 | 0.635 | 0.275 | 105.617 | 1481.806 |
| Wan2.1-FLF2V Wan et al. (2025) | 17.510 | 0.538 | 0.310 | 36.740 | 800.613 | 19.747 | 0.642 | 0.223 | 46.009 | 959.256 |
| TimeLens Tulyakov et al. (2021) | 22.913 | 0.632 | 0.352 | 102.191 | 1706.523 | 27.071 | 0.757 | 0.215 | 79.886 | 1093.200 |
| CBMNet-Large Kim et al. (2023) | 20.633 | 0.742 | 0.343 | 79.459 | 1164.145 | 23.429 | 0.799 | 0.292 | 81.449 | 840.820 |
| TimeLens-XL Ma et al. (2024) | 17.498 | 0.530 | 0.235 | 100.506 | 1438.467 | 26.241 | 0.789 | 0.224 | 82.760 | 557.176 |
| VDM-EVFI-Wan2.1 Chen et al. (2024) | 25.040 | 0.779 | 0.123 | 15.361 | 165.236 | 29.042 | 0.821 | 0.082 | 17.275 | 158.183 |
| Ours | **25.544** | **0.799** | **0.115** | **13.367** | 158.557 | **29.089** | **0.858** | **0.080** | 16.319 | **151.345** |

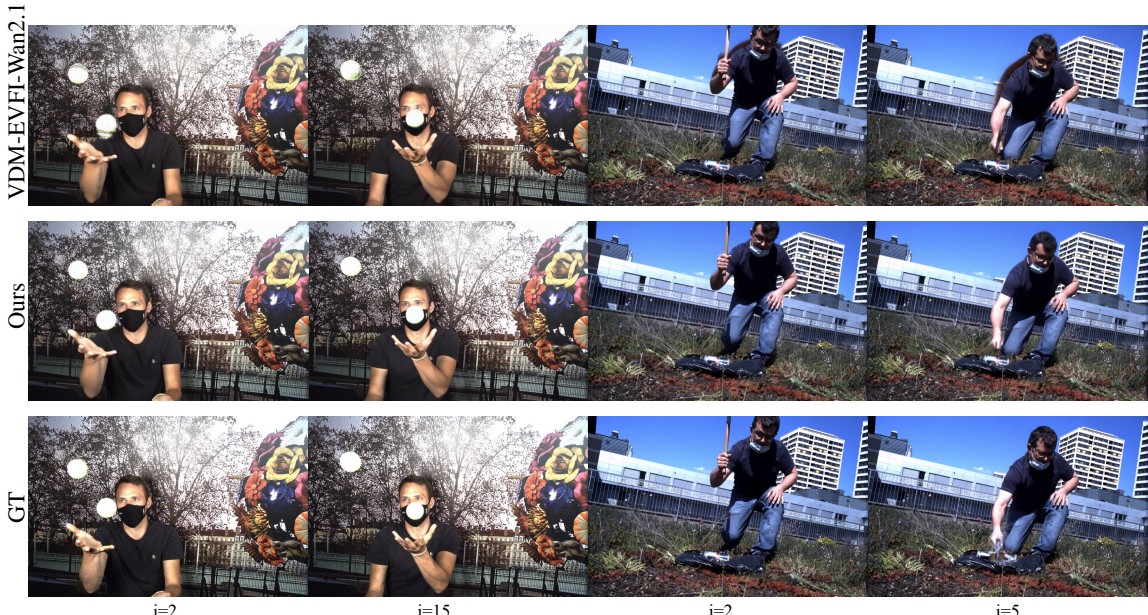

Figure 11: Visual comparison between VDM-EVFI-Wan2.1 and our method on the BS-ERGB dataset.(time × 24).

## B.2 Additional Ablation Study

**Impact of Different Input Features** We evaluate all ablation settings on the BS-ERGB test dataset, and the results are reported in Tab. 7, which indicates that both IWEs and flow warping are benefit for a better performance.

**Impact of Different Injection Blocks** In this work, the Wan2.1 FLF2V backbone contains 40 blocks. We choose the first two blocks as the injection points for our flow-based alignment and fusion module. To further investigate the effect of injection position, we conduct additional ablation studies by inserting the module into the middle two blocks and the last two blocks of the backbone. The corresponding results are reported in Tab. 8. The results indicate that injecting temporal information into the last two blocks yields better LPIPS, FID, and FVD scores, whereas injecting it into the first two blocks leads to better PSNR and SSIM. In this work, we adopt the first two blocks as the injection position to prioritize reconstruction quality.

**Impact of Different Event Representaion** We conducted a literature review on event-based conditioning and identified two representative approaches: the edge-based conditioning method proposed in CUBE Zhao et al. (2024), which converts the event stream into an edge images, and the event voxel stack representation used in VDM-EVFI Chen et al. (2024). We incorporated each of these event representations separately as additional inputs to the Wan2.1 FLF2V by feeding them through an encoder that shares the same architecture as our IWE encoder. The models were trained for 5,400 steps on an NVIDIA A800 GPU and evaluated on the BS-ERGB dataset for fair comparison. The results are summarized in Tab. 9. These results demonstrate that our event representation (IWE and Flow Warping) achieves consistently superior performance across all metrics under the same backbone architecture and training settings.

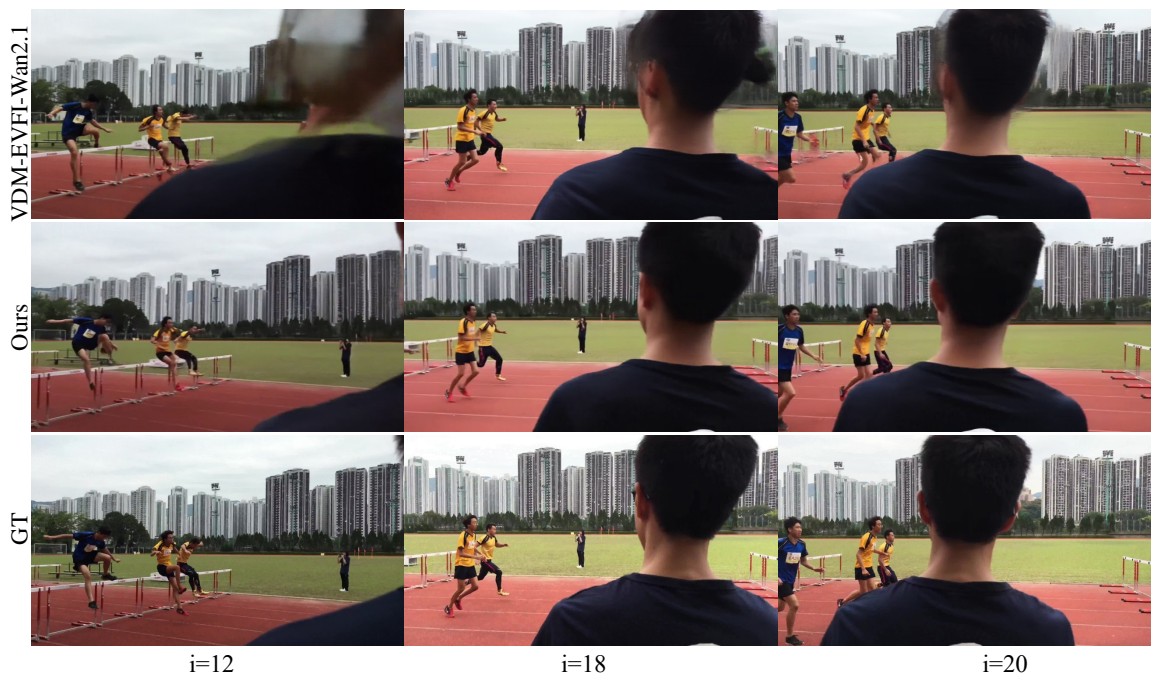

Figure 12: Visual comparison between VDM-EVFI-Wan2.1 and Ours method on the Davis dataset (time $\times$ 24).

Table 7: Ablation study on BS-ERGB test dataset of our method (time $\times$ 24).

| Methods | PSNR↑ | SSIM↑ | LPIPS↓ | FID↓ | FVD↓ |
|---|---|---|---|---|---|
| w/o (IWE and flows) | 17.390 | 0.578 | 0.241 | 22.233 | 678.026 |
| w/o flows warping | 22.751 | 0.685 | 0.126 | 9.207 | 124.715 |
| w/o IWE | 21.237 | 0.626 | 0.189 | 17.807 | 200.861 |
| IWE & flow inputs | 22.839 | 0.689 | 0.126 | 9.402 | 123.306 |
| Full model | **23.072** | **0.693** | **0.126** | **9.023** | **121.620** |

Table 8: Ablation study on BS-ERGB test dataset regarding the injection position of flow-guided blocks (time ×24).

| Injection Point | PSNR↑ | SSIM↑ | LPIPS↓ | FID↓ | FVD↓ |
|---|---|---|---|---|---|
| Last two blocks | 22.814 | 0.691 | **0.120** | **8.986** | **108.093** |
| Mid. two blocks | 22.801 | 0.687 | 0.129 | 9.393 | 121.771 |
| First two blocks | **23.072** | **0.693** | 0.126 | 9.023 | 121.620 |

Table 9: Ablation study on BS-ERGB test dataset of different event conditions (time $\times$ 24).

| Event Rep. | PSNR↑ | SSIM↑ | LPIPS↓ | FID↓ | FVD↓ |
|---|---|---|---|---|---|
| Edge Zhao et al. (2024) | 17.703 | 0.599 | 0.227 | 23.204 | 755.380 |
| Event Voxel Stack Chen et al. (2024) | 21.311 | 0.611 | 0.205 | 18.447 | 228.950 |
| Ours (IWE & Flow) | **23.072** | **0.693** | **0.126** | **9.023** | **121.620** |

## B.3  ADDITIONAL ANALYSIS

### B.3.1  CLARIFICATION ON THE NECESSITY AND ROLE OF EVENT STREAMS

The key advantage of using the raw event stream—rather than external or frame-derived optical flow—lies in its temporal precision, motion observability, and optimization stability.

(1) Fundamental limitation of frame-based optical flow. Frame-based methods (PWC-Net Sun et al. (2018), RIFE Huang et al. (2022), etc.) only observe motion at two discrete time points. As a result, they are fundamentally unable to recover high-frequency or nonlinear motion occurring between the frames. No matter how strong the model is, the intermediate dynamics remain unobserved. This inherent sampling-gap issue explains the inferior performance of RIFE in our quantitative results (Tab. 5).

(2) Events provide dense temporal observations and enable accurate intermediate flow. Events record asynchronous brightness changes at microsecond resolution, giving access to a dense stream of motion cues that frames completely miss. This allows us to reconstruct optical flow and IWEs aligned to any intermediate timestamp, supplying the "missing motion samples" that frame-based flow cannot provide.

(3) Segmented contrast maximization is essential for stable and accurate motion estimation. Using the entire event span to estimate a single IWE leads to structure smear, inconsistent gradients, and a highly non-convex contrast-maximization objective. In contrast, segmenting the event stream into $T$ short intervals (Eq. 6–7) ensures that the displacement within each segment is small and nearly linear, producing sharp IWEs and reliable flow fields (Fig. 13(b)). The full-span IWE (Fig. 13(c)) clearly demonstrates the failure mode—blurred textures and inaccurate motion.

(4) Temporally accurate flow is structurally required by our diffusion pipeline. Our diffusion model warps latent features at multiple intermediate timestamps. Errors of even a few milliseconds accumulate across steps, leading to geometric drift, ghosting, and motion inconsistency. Thus, the temporally coherent, segment-wise flows from events are not only beneficial but necessary for stable latent warping and high-quality interpolation.

(5) Compatibility with diffusion-transformer–based video generation models. Although alternative event representations (e.g., voxel grids, edge/accumulation images) also encode temporal cues, they introduce substantial ambiguity and force the model to implicitly infer complex motion dynamics from high-dimensional, entangled features. In contrast, our event-derived IWEs and optical flows—obtained via contrast maximization—are explicit and physically interpretable descriptors of scene structure and pixel-wise motion. This makes them inherently well aligned with diffusion-transformer architectures, which rely on accurate, structured warping of latent features during the denoising process. Our ablation study (in Tab. 9) corroborates this design choice: substituting IWEs/flows with other event representations markedly increases the learning difficulty and leads to noticeable degradation in interpolation quality. These results demonstrate that physically grounded event-based motion cues provide a far more effective supervisory signal for high-quality frame synthesis.

Together, these points demonstrate that raw event streams provide essential temporal information, enable stable flow estimation, and ensure correct motion propagation in the diffusion model—making them a critical component of our interpolation framework.

### B.3.2  ADDITIONAL DESCRIPTION OF EVPEXELS DATASETS

**Acquisiton method**: We constructed the EvPexels dataset through a two-stage filtering pipeline:

1. Coarse Filtering: We first queried videos from the Pexels website using semantic keywords such as "urban street," "human activity," and other scene-relevant terms to collect a broad set of candidate videos.

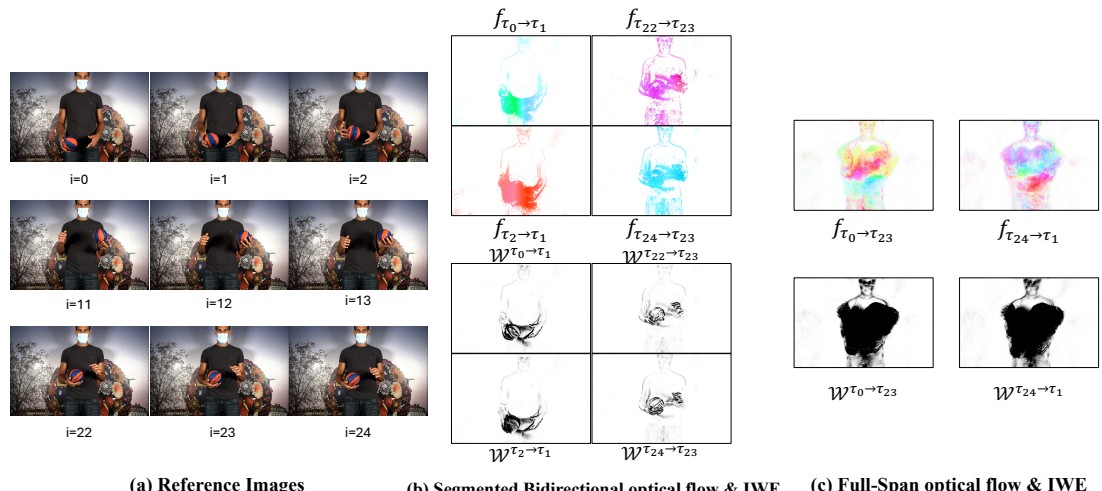

Figure 13: Visual comparison of IWEs and optical flows generated from segmented event intervals versus the full-span event interval.

2. Fine Filtering: Since many downloaded videos contain multiple shots—making them unsuitable for frame interpolation—we applied TransNet V2 Soucek & Lokoc (2024) to detect and retain only single-shot segments.

This pipeline yielded 1,100 high-quality, single-shot videos, each with a resolution of 704 × 480 and no more than 500 frames. Finally, we used the Vid2e simulator Gehrig et al. (2020) to convert each video into a corresponding event stream, resulting in the EvPexels dataset.

**Scene patterns**: The EvPexels dataset exhibits rich scene diversity. Specifically, it contains 223 videos of nature landscape, 198 of forest, 170 of urban street, 119 of beach, and 117 of sports field. Additional common scenes include indoor office (71), gym (49), kitchen (20), playground (16), highway (14), cafe (12), parking lot (11), subway station (11), and shopping mall (10). There are also 59 videos categorized as other. In total, these account for 1,100 videos, reflecting the dataset's broad coverage of real-world environments.

**Motion types**: EvPexels captures a wide spectrum of motion dynamics, essential for evaluating video generation under realistic conditions. The dataset contains 402 videos featuring non-rigid deformation (e.g., cloth waving, human running), 226 videos with complex multi-agent motion (e.g., crowds, team sports), and 170 videos dominated by pure camera motion (pan/tilt/zoom). It also includes 31 videos of rigid object translation (e.g., moving vehicles), alongside examples of rotational motion (e.g., spinning fans or wheels, 12 videos) and oscillatory motion (e.g., bouncing balls or pendulums, 12 videos). The remaining 247 videos belong to other motion categories, further enriching the diversity of temporal dynamics in the dataset.

**Dataset Splits**: For the training phase, we utilize all 1,100 videos contained within the EvPexels dataset. This comprehensive set ensures a robust training foundation covering a wide range of scene types and motion dynamics. For evaluation purposes, instead of using videos from the EvPexels dataset, we have curated an additional set of high-motion videos exclusively from Pexels website. These evaluation videos are carefully selected to ensure there is no overlap with the videos used in the EvPexels training dataset. This approach guarantees an unbiased assessment of model performance on unseen data, accurately reflecting its capability to generalize across different scenes and motion patterns.

