# OpenReview forum: "Effortless Event-Augmented Latent Diffusion for Video Frame Interpolation"
_ICLR.cc/2026/Conference — Submitted to ICLR 2026_

### Official Review · Reviewer_6qaC · 2025-10-28

**Soundness:** 2
**Presentation:** 2
**Contribution:** 2
**Rating:** 4
**Confidence:** 3

**Summary:**

This paper aims to integrate event camera signals into pre-trained Video Latent Diffusion Models for Video Frame Interpolation. This paper proposes a framework that could integrate event signals into pixel-based pre-trained video diffusion models. It consists of an IWE encoder, which encodes IWE into the latent diffusion model, and an Alignment Adapter to align latent features using flows. The IWE and optical flows are obtained from event streams using contrast maximization. The paper also provide a novel synthetic event-video dataset, EvPexels.

**Strengths:**

1. Integrating different form of data representation (event streams) into pre-trained diffusion models is interesting, and the paper successfully addressed the problem.
2. Strong results compared to baseline methods.

**Weaknesses:**

1. Discussion on how event stream helps in this framework. I figure that the proposed method successfully integrates event stream into pre-trained (latent) diffusion models. However, I did not understand how this could greatly help for the task, other than mentioning that there has been prior work that showed the effectiveness of event streams in frame interpolation. This paper does use event streams, but converts them into optical flows and warps the features with those. How would this differ from integrating optical flows, put aside the IWE insertion part? I think discussing the necessity of using event streams could have been very helpful. In the abstract, the authors mention that event camera signals are ideal due to their ability to capture continuous motion at high temporal resolution, but I could not understand where this strength shows its effect in the proposed framework. The ablation study in the Appendix quantitatively shows its advantage, but I believe it is also very important to explain theoretically how/where/why the proposed design is useful, along with the limits / reason for failure of existing methods in comparison.
2. Novelty. To begin with, I really like the approach and found it interesting. However, when it comes to dissection of the proposed components, I found it a little weak in terms of technical novelty. 1) Converting event streams to IWE and flows rely on contrast maximization, an existing method. 2) IWE encoder is a sequence of 3D convolutional layers, to encode and inject IWE signals. 3) Alignment Adapter sounds quite similar to feature warping and aligning which have been conducted in the field of VFI [1,2,3]. 4) Adapting LoRA into training for efficient tuning is a commonly used technique, especially when it comes to adapting a pre-trained model for different purposes [4], as in this paper’s scenario. Integrating existing techniques appropriately can be considered a contribution, but with respect, one could find it a little weak in terms of technical novelty. In case of any possible misunderstandings I have made, please point me out.

My main reason for the rating is W1, on discussing why this problem is important, and theoretical explanation / analysis on how their approach helps.

[1] Niklaus, Simon, and Feng Liu. "Context-aware synthesis for video frame interpolation." CVPR 2018

[2] Niklaus, Simon, and Feng Liu. "Softmax splatting for video frame interpolation." CVPR 2020

[3] Sim, H., Oh, J., and Kim, M. "Xvfi: extreme video frame interpolation." ICCV 2021

[4] Zhang, Lvmin, Anyi Rao, and Maneesh Agrawala. "Adding conditional control to text-to-image diffusion models." ICCV 2023.

**Questions:**

1. I did not quite understand how you actually add intermediate frames. Like, you get N frame latents, and how do you add the N x 23 intermediate frames’ latents to be generated? Is it simply adding Nx23xHxW tokens in between the existing frame tokens?

---

> ### Author Response · Authors · 2025-11-24
> **Response to Reviewer 6qaC (Part 1)**
>
> **We sincerely thank the reviewer for acknowledging that our event representation approach is interesting and for recognizing the strength of our results. We address the reviewer’s concerns and suggestions below and will incorporate the corresponding revisions into the final version of the submission. We have also updated the submitted PDF by appending a Rebuttal Appendix as the final section of the manuscript, and we kindly invite the reviewer to consult this additional material.**
>
>  ---
>
> ### **1: Clarification on the Necessity and Role of Event Streams**
> The key advantage of using the raw event stream, instead of external or frame-derived optical flow, lies in its temporal precision, motion observability, and optimization stability.
>
> (1) Fundamental limitation of frame-based optical flow.
> Frame-based methods (PWC-Net, RIFE, etc.) only observe motion at two discrete time points. As a result, they are fundamentally unable to recover high-frequency or nonlinear motion occurring between the frames. No matter how strong the model is, the intermediate dynamics remain unobserved. This inherent sampling-gap issue explains the inferior performance of RIFE in our quantitative results (Tab.5).
>
> (2) Events provide dense temporal observations and enable accurate intermediate flow.
> Events record asynchronous brightness changes at microsecond resolution, giving access to a dense stream of motion cues that frames completely miss. This allows us to reconstruct optical flow and IWEs aligned to any intermediate timestamp, supplying the “missing motion samples” that frame-based flow cannot provide.
>
> (3) Segmented contrast maximization is essential for stable and accurate motion estimation.
> Using the entire event span to estimate a single IWE leads to structure smear, inconsistent gradients, and a highly non-convex contrast-maximization objective. In contrast, segmenting the event stream into
> $T$ short intervals (Eq.~6–7) ensures that the displacement within each segment is small and nearly linear, producing sharp IWEs and reliable flow fields (Fig.13(b)).
> The full-span IWE (Fig. 13(c)) clearly demonstrates the failure mode, with blurred textures and inaccurate motion.
>
> (4) Temporally accurate flow is structurally required by our diffusion pipeline.
> Our diffusion model warps latent features at multiple intermediate timestamps. Errors of even a few milliseconds accumulate across steps, leading to geometric drift, ghosting, and motion inconsistency. Thus, the temporally coherent, segment-wise flows from events are not only beneficial but necessary for stable latent warping and high-quality interpolation.
>
> (5) Compatibility with diffusion-transformer–based video generation models.
> Although alternative event representations (e.g., voxel grids, edge/accumulation images) also encode temporal cues, they introduce substantial ambiguity and force the model to implicitly infer complex motion dynamics from entangled features. In contrast, our event-derived IWEs and optical flows, obtained via contrast maximization, are explicit and physically interpretable descriptors of scene structure and pixel-wise motion. This makes them naturally aligned with diffusion-transformer architectures, which rely on accurate and structured warping of latent features during the denoising process.
> Our ablation study (in Tab.9) corroborates this design choice: substituting IWEs&flows with other event representations markedly increases the learning difficulty and leads to noticeable degradation in interpolation quality. These results demonstrate that physically grounded event-based motion cues provide a far more effective supervisory signal for high-quality frame synthesis.
>
> Together, these points demonstrate that raw event streams provide essential temporal information, enable stable flow estimation, and ensure correct motion propagation in the diffusion model—making them a critical component of our interpolation framework.
> Please refer to the Appendix B.3.1 for details.
>
> ---
>
> ### **2: Novelty clarification regarding DiT-based video generation models**
> To the best of our knowledge, our work is the first to incorporate event-driven, physically interpretable optical flow into a DiT-based video generation/interpolation framework. Existing diffusion-transformer approaches do not employ explicit motion warping in the latent space, let alone event-derived flow.
> Meanwhile, previous flow-guided video synthesis methods rely on frame-based CNN/UNet architectures and external flow estimators, and are fundamentally different from our event-driven, contrast-maximized, segment-wise motion modeling.
>
> Thus, our formulation introduces a new paradigm:
> integrating event-based motion cues with latent-space warping inside a diffusion transformer,
> which has not been explored in prior work and constitutes a core novelty of our method.

---

> ### Author Response · Authors · 2025-11-24
> **Response to Reviewer 6qaC (Part 2)**
>
> ### **3: How we add intermediate frames**
>
> We do not insert any additional tokens into the DiT sequence.
>
> In the Wan2.1 FLF2V architecture, the DiT always produces a fixed-length latent video of 81 frames.
> The video VAE then reduces the temporal resolution by a factor of 4, resulting in 21 latent frames
> (⌈81 / 4⌉ = 21). This latent sequence length is fixed and independent of the interpolation ratio.
>
> To obtain the desired 24 output frames (i.e., ×24 interpolation), we simply sample the corresponding timestamps
> from this fixed 81-frame temporal grid (or equivalently from its 21-frame latent representation).
> Thus, the intermediate frames are generated by temporal resampling, not by extending the token sequence.
>
> During diffusion, each DiT block receives latent features of shape [B, Seq_Len, C].
> Before each block, we reshape the latent representation into [B, C, L, H, W] with L = 21,
> apply our event-driven flow-warping module to obtain a residual of the same shape,
> and add it to the latent features.
> The updated latent is then reshaped back to [B, Seq_Len, C] and passed to the next DiT block.
> This operation modifies the latent features but does not change the number of latent frames or tokens.
>
> The full code will be released for reproducibility.

---

### Official Review · Reviewer_uM48 · 2025-10-28

**Soundness:** 2
**Presentation:** 3
**Contribution:** 3
**Rating:** 6
**Confidence:** 4

**Summary:**

This paper addresses video frame interpolation (VFI) using latent diffusion models, focusing on challenges like large temporal gaps and complex motion that cause artifacts. The authors propose integrating event camera signals, which capture continuous high-temporal-resolution motion, into a pre-trained Diffusion Transformer (DiT)-based video model. Their method extracts Image Warped Events (IWEs) and bidirectional sparse optical flow from event streams via contrast maximization, then injects these cues using lightweight adapters: an IWE encoder for spatial structure and a flow-based alignment-and-fusion adapter for temporal consistency. A synthetic event-video dataset (EvPexels) with 1,100 scenes is introduced. Experiments on real and synthetic datasets show improved interpolation quality over state-of-the-art methods in metrics like LPIPS, FID, and FVD.

**Strengths:**

-The motivation of incorporating event-derived signals (IWEs, optical flow) into a pre-trained DiT model via lightweight adapters is well explained. This is reasonable because the two extracted pieces of information directly correspond to the challenges in previous VFI methods (large temporal gaps and complex motion that cause artifacts).

-The proposed EvPexels dataset, with 1100 diverse motion-rich scenes, addresses the scarcity of large-scale paired event-video data and supports training and benchmarking. If the dataset and tools will be released, it will promote future research in event-driven video generation.

**Weaknesses:**

-I believe the paper lacks experiments that directly compare the proposed motion information extraction with other event‑based conditioning methods under the same baseline. I trust that the ablations in Table 4 show that the designed conditions work, but it is still necessary to explain their advantages over other comparable approaches to further justify the idea.

-The EvPexels dataset is introduced but lacks detailed analysis of its motion diversity, scene types, and potential biases, which are crucial for assessing generalization. Then, the dataset split (training/test) for EvPexels is not specified, raising concerns about evaluation fairness and overfitting. Since the authors present the data as a contribution, they should conduct a comprehensive analysis and evaluation of the proposed new dataset to demonstrate that it indeed has value for advancing research in this field.

**Questions:**

-I recommend using the experimental setting in Table 4 as the baseline and attempting to use other event based conditioning to guide DiT for VFI, comparing against the proposed approach. I also suggest that the authors provide a more detailed review of conditioning methods that may work, in order to highlight the originality of the proposed design.

-I hope the authors can provide more details about EvPexels, including the acquisition method, motion patterns, scene types, and dataset splits. It would be preferable to also include analyses showing why this dataset is important to the field and in what aspects it enables evaluation or training that other datasets cannot.

---

> ### Author Response · Authors · 2025-11-24
> **Response to Reviewer uM48 (Part 1)**
>
> **We thank the reviewer for acknowledging that the motivation of our work is well explained, that the proposed method is reasonable, and for recognizing the contribution of the EvPexels dataset. We address the reviewer’s concerns and suggestions below and will incorporate the corresponding revisions into the final version of the submission. We have also updated the submitted PDF by appending a Rebuttal Appendix as the final section of the manuscript, and we kindly invite the reviewer to consult this additional material.**
>
> ---
>
> ### **1: Additional event‑based conditionings**
> Thank you for your suggestion. We conducted a literature review on event-based conditioning and identified two representative approaches: the edge-based conditioning method proposed in CUBE (Zhao et al., 2024), which converts the event stream into edge images, and the event voxel stack representation used in VDM-EVFI (Chen et al., 2024).
>
> We incorporated each of these event representations separately as additional inputs to the Wan2.1 FLF2V by feeding them through an encoder that shares the same architecture as our IWE encoder. The models were trained for 5,400 steps on an NVIDIA A800 GPU and evaluated on the BS-ERGB dataset for fair comparison. The results are summarized below.
>
> | **Event Representation**                                      | **PSNR↑** | **SSIM↑** | **LPIPS↓** | **FID↓** | **FVD↓**  |
> |----------------------------------------------------|-----------|-----------|------------|----------|-----------|
> | Edge (Zhao et al. 2024)                                           | 17.703    | 0.599     | 0.227      | 23.204   | 755.380   |
> | Event Voxel Stack (Chen at al. 2024)                              | 21.311    | 0.611     | 0.205      | 18.447   | 228.950   |
> | Ours (IWE & Flow Warping)                                  | **23.072**    | **0.693**     | **0.126**      | **9.023**    | **121.620**   |
>
> These results demonstrate that our event representation (IWE & Flow Warping) achieves consistently superior performance across all metrics under the same backbone architecture and training settings.
>
>
> **References**:
>
> Zhao et al. Controllable unsupervised event-based video generation. In 2024 IEEE International Conference on Image Processing (ICIP). 2024
>
> Chen et al. Repurposing pre-trained video diffusion models for
> event-based video interpolation (CVPR). 2025

---

> ### Author Response · Authors · 2025-11-24
> **Response to Reviewer uM48 (Part 2)**
>
> ### **2: Description of the EvPexels Dataset**
>
> **1: Acquisiton method:**
> We constructed the EvPexels dataset through a two-stage filtering pipeline:
>
> - A: Coarse Filtering: We first queried videos from the Pexels website using semantic keywords such as “urban street,” “human activity,” and other scene-relevant terms to collect a broad set of candidate videos.
>
> - B: Fine Filtering: Since many downloaded videos contain multiple shots—making them unsuitable for frame interpolation—we applied TransNet V2 (Soucek & Lokoc, 2024) to detect and retain only single-shot segments.
>
> This pipeline yielded 1,100 high-quality, single-shot videos, each with a resolution of 704 × 480 and no more than 500 frames. Finally, we used the Vid2e simulator (Gehrig et al., 2020) to convert each video into a corresponding event stream, resulting in the EvPexels dataset.
>
> **2: Scene patterns:**
> The EvPexels dataset exhibits rich scene diversity. Specifically, it contains 223 videos of nature landscape, 198 of forest, 170 of urban street, 119 of beach, and 117 of sports field. Additional common scenes include indoor office (71), gym (49), kitchen (20), playground (16), highway (14), cafe (12), parking lot (11), subway station (11), and shopping mall (10). There are also 59 videos categorized as other. In total, these account for 1,100 videos, reflecting the dataset’s broad coverage of real-world environments.
>
> **3：Motion types:**
> EvPexels captures a wide spectrum of motion dynamics, essential for evaluating video generation under realistic conditions. The dataset contains 402 videos featuring non-rigid deformation (e.g., cloth waving, human running), 226 videos with complex multi-agent motion (e.g., crowds, team sports), and 170 videos dominated by pure camera motion (pan/tilt/zoom). It also includes 31 videos of rigid object translation (e.g., moving vehicles), alongside examples of rotational motion (e.g., spinning fans or wheels, 12 videos) and oscillatory motion (e.g., bouncing balls or pendulums, 12 videos). The remaining 247 videos belong to other motion categories, further enriching the diversity of temporal dynamics in the dataset.
>
> **4: Dataset Splits:**
> For the training phase, we utilize all 1,100 videos contained within the EvPexels dataset. This comprehensive set ensures a robust training foundation covering a wide range of scene types and motion dynamics.
> For evaluation purposes, instead of using videos from the EvPexels dataset, we have curated an additional set of high-motion videos exclusively from Pexels website. These evaluation videos are carefully selected to ensure there is no overlap with the videos used in the EvPexels training dataset. This approach guarantees an unbiased assessment of model performance on unseen data, accurately reflecting its capability to generalize across different scenes and motion patterns.
>
> The EvPexels dataset will be publicly released upon publication, ensuring full reproducibility and enabling further research.

---

### Official Review · Reviewer_9sZG · 2025-10-30

**Soundness:** 3
**Presentation:** 3
**Contribution:** 2
**Rating:** 4
**Confidence:** 3

**Summary:**

This paper addresses video frame interpolation using event camera signals as control. It extracts both the IWE representation and optical flow from the event data. The IWE representation is concatenated with the noised video latents and fed into the diffusion denoising network, while the optical flow is used to warp features of neighboring frames, aligning them with the original DiT features. This design enables the model to incorporate explicit motion guidance for more accurate interpolation.

**Strengths:**

The paper tackles a well-defined problem—event-based video frame interpolation. Compared to earlier event based video frame interpolation,  this work uses both IWE representation and optical flow extracted from the event signal to adapt to the  video inbetweening diffusion model.

**Weaknesses:**

Compared to prior work, such as VDM-EVFI (Chen et al., 2024), the proposed method adopts a different video diffusion backbone and a new strategy for incorporating event representations into the network.

To fully demonstrate the effectiveness of the proposed approach, it would be fair to include a comparison with a version of VDM-EVFI adapted to the same video diffusion model (Wan 2.1 FLF2V) used in this paper. Without such a comparison, it is difficult to determine whether the performance improvement over VDM-EVFI stems from the new event representation or simply from using a stronger diffusion backbone.

I am also confused about the evaluate setting in the paper, please see Questions section.

**Questions:**

1. I did not fully understand why the method is evaluated on datasets such as DAVIS and Pexels, which do not contain event signals for interpolation. What is the evaluation setting when comparing on these datasets? Why not focus solely on event-based VFI datasets instead?
2.  It is also unclear why the ablation studies are conducted on Pexels and DAVIS rather than on BS_ERGB, which is an event-based interpolation dataset.
3. Which layers in the DiT architecture use the alignment adapter with optical flow? Did the authors conduct any ablation studies on this component?

---

> ### Author Response · Authors · 2025-11-24
> **Response to Reviewer 9sZG (Part 1)**
>
> **We thank the reviewer for acknowledging that the design of our method enables accurate interpolation. We address the reviewer’s concerns and suggestions below and will incorporate the corresponding revisions into the final version of the submission. We have also updated the submitted PDF by appending a Rebuttal Appendix as the final section of the manuscript, and we kindly invite the reviewer to consult this additional material.**
>
> ---
>
> ### **1: Fair comparison**
>
> To address the concern regarding unfair comparison, we adapt the VDM-EVFI method built on Stable Video Diffusion to the Wan2.1 FLF2V backbone, which we denote as VDM-EVFI-Wan2.1. We retrain this model from scratch on our training set. For fairness, the training configuration strictly matches ours: we use the same 8 NVIDIA A800 GPUs and maintain the same number of training iterations.
>
> We evaluate VDM-EVFI-Wan2.1 on the BS-ERGB, DAVIS, and Pexels datasets. The results are summarized below, with additional details and visual comparisons provided in Appendix B.1. The quantitative comparison shows that VDM-EVFI-Wan2.1 remains inferior to our method even when using the same backbone, clearly demonstrating the effectiveness of our design.
>
> **Table 5.** Quantitative comparison of VFI performance on the BS-ERGB test set (×24 interpolation). **Bold indicates the best performance.**
> | **Methods** | PSNR↑ | SSIM↑ | LPIPS↓ | FID↓ | FVD↓ |
> |------------|-------|--------|---------|--------|---------|
> | RIFE   | 22.174 | 0.641 | 0.172 | 35.347 | 1113.496 |
> | TRF   | 14.078 | 0.4117 | 0.426 | 47.146 | 971.424 |
> | GI   | 16.964 | 0.518 | 0.311 | 33.082 | 588.371 |
> | ViBiD  | 15.525 | 0.475 | 0.352 | 39.027 | 788.652 |
> | FCVG   | 17.809 | 0.546 | 0.302 | 26.832 | 726.752 |
> | Wan2.1-FLF2V  | 18.698 | 0.618 | 0.212 | 18.607 | 376.828 |
> | TimeLens  | 24.704 | 0.699 | 0.165 | 43.808 | 851.523 |
> | CBMNet-Large | **25.306** | **0.712** | 0.169 | 17.658 | 228.753 |
> | TimeLens-XL  | 21.737 | 0.678 | 0.248 | 47.155 | 710.688 |
> | VDM-EVFI-Wan2.1 | 22.402 | 0.673 | 0.282 | 15.693 | 145.067 |
> | **Ours** | 23.261 | 0.704 | **0.132** | **8.168** | **117.368** |
>
> ---
>
>
>
> **Table 6.** Quantitative comparison on DAVIS and Pexels datasets (x24 interpolation).
> | **Methods** | PSNR↑ | SSIM↑ | LPIPS↓ | FID↓ | FVD↓ | PSNR↑ | SSIM↑ | LPIPS↓ | FID↓ | FVD↓ |
> |-------------|-------|--------|---------|-------|--------|--------|--------|---------|--------|---------|
> | | **DAVIS** | | | | | **Pexels** | | | | |
> | RIFE | 18.287 | 0.487 | 0.402 | 82.137 | 2028.480 | 21.820 | 0.634 | 0.274 | 80.118 | 1976.327 |
> | TRF | 14.132 | 0.459 | 0.484 | 70.528 | 1373.954 | 16.737 | 0.600 | 0.400 | 109.516 | 1624.791 |
> | GI | 14.850 | 0.467 | 0.406 | 55.067 | 1158.330 | 17.700 | 0.600 | 0.306 | 109.029 | 1212.097 |
> | ViBiD | 14.811 | 0.456 | 0.448 | 55.343 | 1194.670 | 17.413 | 0.588 | 0.365 | 104.089 | 1335.211 |
> | FCVG | 16.162 | 0.509 | 0.385 | 48.839 | 1246.823 | 19.172 | 0.635 | 0.275 | 105.617 | 1481.806 |
> | Wan2.1-FLF2V | 17.510 | 0.538 | 0.310 | 36.740 | 800.613 | 19.747 | 0.642 | 0.223 | 46.009 | 959.256 |
> | TimeLens | 22.913 | 0.632 | 0.352 | 102.191 | 1706.523 | 27.071 | 0.757 | 0.215 | 79.886 | 1093.200 |
> | CBMNet-Large | 20.633 | 0.742 | 0.343 | 79.459 | 1164.145 | 23.429 | 0.799 | 0.292 | 81.449 | 840.820 |
> | TimeLens-XL | 17.498 | 0.530 | 0.235 | 100.506 | 1438.467 | 26.241 | 0.789 | 0.224 | 82.760 | 557.176 |
> | VDM-EVFI-Wan2.1 | 25.040 | 0.779 | 0.123 | 15.361 | 165.236 | 29.042 | 0.821 | 0.082 | 17.275 | 158.183 |
> | **Ours** | **25.544** | **0.799** | **0.115** | **13.367** | **158.557** | **29.089** | **0.858** | **0.080** | **16.319** | **151.345** |
>
> ---
>
> ### **2: Evaluated on the Davis and Pexels**
> We evaluate on DAVIS and Pexels for two reasons.
> First, following prior event-guided VFI works (TimeLens, CBMNet, TimeLens-XL), it is standard to evaluate on both real-event and synthetic-event datasets to assess a model’s generalization. Real event datasets are very limited in scale, motion diversity, and scene coverage, making them insufficient for measuring the robustness of event-guided methods.
>
> Second, although DAVIS and Pexels do not contain real events, we use the standard synthetic-event evaluation protocol adopted in prior works, where events are generated using the v2e simulator (Hu et al. 2021). This produces event–video pairs that allow all methods to be compared under the same ×24 interpolation setting. For methods that do not support ×24 interpolation (e.g., VDM-EVFI), we evaluate them with their original ×12 setting.
>
> Therefore, evaluating on DAVIS and Pexels is not only consistent with the event-VFI literature but also essential for testing generalization across a wider range of motions and scenes, beyond what real-event datasets alone can provide.
>
> [1] Hu et al. v2e: From Video Frames to Realistic DVS Events. In 2021 IEEE/CVF Conference on Computer Vision and Pattern Recognition Workshops (CVPRW)

---

> ### Author Response · Authors · 2025-11-24
> **Response to Reviewer 9sZG (Part 2)**
>
> ### **3:  Ablation Study on the Real-Event Dataset**
> We thank the reviewer for the helpful suggestion. The additional ablation study performed on the BS-ERGB test dataset under the ×24 interpolation setting is summarized in the table below. Further details can be found in the Rebuttal Appendix (Appendix B.2: Additional Ablation Study).
>
> **Table 7.** Ablation study on BS-ERGB test dataset of our method (time ×24)
>
> | **Methods**             | PSNR↑  | SSIM↑ | LPIPS↓ | FID↓   | FVD↓     |
> |-------------------------|--------|--------|---------|---------|-----------|
> | w/o (IWE and flows)     | 17.390 | 0.578 | 0.241   | 22.233  | 678.026   |
> | w/o flows warping       | 22.751 | 0.685 | 0.126   | 9.207   | 124.715   |
> | w/o IWE                 | 21.237 | 0.626 | 0.189   | 17.807  | 200.861   |
> | IWE & flow inputs       | 22.839 | 0.689 | 0.126   | 9.402   | 123.306   |
> | **Full model**          | **23.072** | **0.693** | **0.126** | **9.023** | **121.620** |
>
> ---
>
> ### **4: Ablation Study on Block Layer Selection**
> We thank the reviewer for the helpful suggestion. In this ablation study on the BS-ERGB test dataset under the ×24 interpolation setting, we investigate the effect of different injection positions for the flow-guided blocks within the Wan2.1 FLF2V backbone, which consists of 40 blocks in total. Specifically, we apply the flow-based alignment and fusion modules to the first two blocks, the middle two blocks, and the last two blocks, respectively. The quantitative results are presented below.
>
> **Table 8.** Ablation study on BS-ERGB test dataset regarding the injection position of flow-guided blocks (time ×24)
> | **Injection Point**   | PSNR↑  | SSIM↑ | LPIPS↓ | FID↓  | FVD↓    |
> |-----------------------|--------|--------|---------|--------|----------|
> | Last two blocks       | 22.814 | 0.691 | **0.120** | **8.986** | **108.093** |
> | Mid. two blocks       | 22.801 | 0.687 | 0.129   | 9.393  | 121.771  |
> | First two blocks      | **23.072** | **0.693** | 0.126   | 9.023  | 121.620  |
>
> The results indicate that injecting temporal information into the last two blocks yields better LPIPS, FID, and FVD scores, while injecting into the first two blocks provides better PSNR and SSIM. In this work, we choose the **first two blocks** as the injection position to prioritize reconstruction quality. Further details can be found in the Rebuttal Appendix (Appendix B.2: Additional Ablation Study).

---

### Official Review · Reviewer_BPER · 2025-10-31

**Soundness:** 2
**Presentation:** 2
**Contribution:** 2
**Rating:** 4
**Confidence:** 3

**Summary:**

The paper proposes an adapter-based framework that integrates event camera signals into a pre-trained DiT-based latent diffusion model for video frame interpolation. The authors extract Image Warped Events (IWEs) and bidirectional sparse optical flow via contrast maximization and inject them into the latent space through two lightweight adapters: an IWE encoder for spatial structure and an alignment-and-fusion module for temporal consistency. Experimental results on several datasets (BS-ERGB, DAVIS, Pexels) show strong perceptual metrics and competitive reconstruction accuracy compared to prior diffusion-based and event-guided methods.

**Strengths:**

- The authors explore to inject event signal into pre-trained DiT-based model for video frame interpolation.
- The introduction of the EvPexels dataset could be a useful contribution for benchmarking event-based video tasks.
- The proposed method outperforms the baselines and achieves competitive visual results according to the experiments provided.

**Weaknesses:**

- Unfair comparison with prior event-based baselines. Most previous event-based generative methods, such as *VDM-EVFI*, are built upon the U-Net–based *Stable Video Diffusion (SVD)* backbone, whereas the proposed approach leverages the more advanced *Wan2.1* DiT architecture. This discrepancy makes the direct comparison somewhat unfair, as it is difficult to disentangle the performance improvements brought by the proposed event integration from those inherited from the stronger base model.
- Limited qualitative comparisons. The paper provides only a single qualitative example comparing the proposed method with prior works in Figure 3, and notably omits key baselines such as *VDM-EVFI*. Additional qualitative results would be valuable for demonstrating the visual benefits of the proposed method, especially under challenging motion scenarios.
- Lack of comparison with non-generative VFI methods. While traditional non-generative interpolation models may struggle with large temporal gaps, including one or two representative non-generative baselines (e.g., those used in *VDM-EVFI*) would help clarify the practical advantages and trade-offs of the proposed generative approach, making the evaluation more comprehensive and contextually grounded.

**Questions:**

- How do the authors ensure a fair comparison given that most event-based baselines (e.g., *VDM-EVFI*) are built upon the U-Net–based *Stable Video Diffusion (SVD)*, while the proposed method uses the more powerful *Wan2.1* DiT architecture?
- Have the authors conducted any ablation or control experiment using the same backbone (e.g., Wan2.1 without event integration) to isolate the contribution of the proposed event adapters?
- Why do the qualitative comparisons include only one example, and why is *VDM-EVFI*—a closely related event-guided diffusion baseline—excluded from the visual comparisons?
- Could the authors provide additional qualitative results on diverse or challenging motion scenes (e.g., occlusion, or high-speed motion) to better demonstrate the strengths and limitations of the proposed method?
- Have the authors considered including one or two representative non-generative VFI baselines for completeness, as the VDM-EVFI did?

---

> ### Author Response · Authors · 2025-11-24
> **Response to Reviewer BPER (Part 1)**
>
> **We thank the reviewer for acknowledging the competitive visual performance of our method and recognizing the potential value of the proposed EvPexels dataset. We address the reviewer’s concerns and suggestions below and will incorporate the corresponding revisions into the final version of the submission. We have also updated the submitted PDF by appending a Rebuttal Appendix as the final section of the manuscript, and we kindly invite the reviewer to consult this additional material.**
>
> ---
>
> ### **1: Fair comparison and comparison with non-generative VFI methods**
>
> To address the concern regarding unfair comparison, we adapt the VDM-EVFI method built on Stable Video Diffusion to the Wan2.1 FLF2V backbone, which we denote as **VDM-EVFI-Wan2.1**. We retrain this model from scratch on our training set. For fairness, the training configuration strictly matches ours: we use the 8 NVIDIA A800 GPUs and keep the same number of training iterations.
>
> Regarding the request for non-generative baselines, following the evaluation protocol of VDM-EVFI, we additionally include the non-generative baseline **RIFE** for quantitative comparison.
>
> We evaluate both VDM-EVFI-Wan2.1 and RIFE on the BS-ERGB, DAVIS, and Pexels datasets. The results are summarized below, with additional details and visual comparisons provided in Appendix B.1.
> The quantitative comparison shows that **VDM-EVFI-Wan2.1 remains inferior to our method even with the same backbone**, and non-generative baseline RIFE is limited in large-motion scenarios.
>
> **Table 5.** Quantitative comparison of VFI performance on the BS-ERGB test set (×24 interpolation).
> | **Methods** | PSNR↑ | SSIM↑ | LPIPS↓ | FID↓ | FVD↓ |
> |------------|-------|--------|---------|--------|---------|
> | RIFE   | 22.174 | 0.641 | 0.172 | 35.347 | 1113.496 |
> | TRF   | 14.078 | 0.4117 | 0.426 | 47.146 | 971.424 |
> | GI   | 16.964 | 0.518 | 0.311 | 33.082 | 588.371 |
> | ViBiD  | 15.525 | 0.475 | 0.352 | 39.027 | 788.652 |
> | FCVG   | 17.809 | 0.546 | 0.302 | 26.832 | 726.752 |
> | Wan2.1-FLF2V  | 18.698 | 0.618 | 0.212 | 18.607 | 376.828 |
> | TimeLens  | 24.704 | 0.699 | 0.165 | 43.808 | 851.523 |
> | CBMNet-Large | **25.306** | **0.712** | 0.169 | 17.658 | 228.753 |
> | TimeLens-XL  | 21.737 | 0.678 | 0.248 | 47.155 | 710.688 |
> | VDM-EVFI-Wan2.1 | 22.402 | 0.673 | 0.282 | 15.693 | 145.067 |
> | **Ours** | 23.261 | 0.704 | **0.132** | **8.168** | **117.368** |
>
> ---
>
> **Table 6.** Quantitative comparison on DAVIS and Pexels datasets (x24 interpolation).
> | **Methods** | PSNR↑ | SSIM↑ | LPIPS↓ | FID↓ | FVD↓ | PSNR↑ | SSIM↑ | LPIPS↓ | FID↓ | FVD↓ |
> |-------------|-------|--------|---------|-------|--------|--------|--------|---------|--------|---------|
> | | **DAVIS** | | | | | **Pexels** | | | | |
> | RIFE | 18.287 | 0.487 | 0.402 | 82.137 | 2028.480 | 21.820 | 0.634 | 0.274 | 80.118 | 1976.327 |
> | TRF | 14.132 | 0.459 | 0.484 | 70.528 | 1373.954 | 16.737 | 0.600 | 0.400 | 109.516 | 1624.791 |
> | GI | 14.850 | 0.467 | 0.406 | 55.067 | 1158.330 | 17.700 | 0.600 | 0.306 | 109.029 | 1212.097 |
> | ViBiD | 14.811 | 0.456 | 0.448 | 55.343 | 1194.670 | 17.413 | 0.588 | 0.365 | 104.089 | 1335.211 |
> | FCVG | 16.162 | 0.509 | 0.385 | 48.839 | 1246.823 | 19.172 | 0.635 | 0.275 | 105.617 | 1481.806 |
> | Wan2.1-FLF2V | 17.510 | 0.538 | 0.310 | 36.740 | 800.613 | 19.747 | 0.642 | 0.223 | 46.009 | 959.256 |
> | TimeLens | 22.913 | 0.632 | 0.352 | 102.191 | 1706.523 | 27.071 | 0.757 | 0.215 | 79.886 | 1093.200 |
> | CBMNet-Large | 20.633 | 0.742 | 0.343 | 79.459 | 1164.145 | 23.429 | 0.799 | 0.292 | 81.449 | 840.820 |
> | TimeLens-XL | 17.498 | 0.530 | 0.235 | 100.506 | 1438.467 | 26.241 | 0.789 | 0.224 | 82.760 | 557.176 |
> | VDM-EVFI-Wan2.1 | 25.040 | 0.779 | 0.123 | 15.361 | 165.236 | 29.042 | 0.821 | 0.082 | 17.275 | 158.183 |
> | **Ours** | **25.544** | **0.799** | **0.115** | **13.367** | **158.557** | **29.089** | **0.858** | **0.080** | **16.319** | **151.345** |
>
> ---
>
>
> ### **2: Qualitative comparisons**
>
> Thank you for highlighting the importance of qualitative evaluation. Our originally submitted paper already includes extensive qualitative comparisons beyond the single example shown in Fig. 3.
> - Appendix A (Fig. 5) presents a camera-motion scene under ×24 interpolation across multiple methods.
> - Fig. 6 and 7 in Appendix A compare our approach with Wan2.1 FLF2V under large-motion scenarios.
> - Appendix A.2.3 illustrates challenging occlusion and large-motion cases (Figs. 8 and 9), where VDM-EVFI is explicitly included as a baseline.
> - The supplementary video provides video-based qualitative comparisons, including high-speed sports sequences.
>
> Together, these qualitative results further demonstrate the advantages of our method and complement the representative example shown in the main paper.

---

> ### Author Response · Authors · 2025-11-24
> **Response to Reviewer BPER (Part 2)**
>
> ### **3: Clarification on Ablation Using the Same Backbone**
>
> In the originally submitted paper, we conducted an ablation study using the same Wan2.1 FLF2V backbone without any event conditions. We also evaluated configurations including:
>
> - removing IWE,
> - removing the flow-warping module,
> - using IWE + optical flow as input (without warping).
>
> The corresponding results and analyses are provided in Tab. 4 of Appendix A.1.1. These experiments consistently demonstrate that incorporating event information leads to improved interpolation performance even when the backbone remains unchanged. For clarity, we also reproduce the ablation results here.
>
>
> **Table 4.** Ablation study on Pexels test dataset of our method (time ×24)
>
> | **Method** | PSNR↑ | SSIM↑ | LPIPS↓ | FID↓ | FVD↓ |
> |------------|--------|--------|----------|---------|---------|
> | w/o (IWE & flows) | 16.928 | 0.577 | 0.301 | 59.362 | 1823.590 |
> | w/o flows warping | 25.130 | 0.783 | 0.110 | 27.207 | 252.806 |
> | w/o IWE | 25.231 | 0.785 | 0.111 | 27.249 | 242.496 |
> | IWE & flow inputs | 25.179 | 0.790 | 0.112 | 27.519 | 234.407 |
> | **Full model** | **25.693** | **0.804** | **0.100** | **23.445** | **217.054** |

---

### Author Response · Authors · 2025-11-28
**Gentle Reminder: Please Review Our Rebuttal Submission**

Dear Reviewers and Area Chair,

We would like to kindly remind the reviewers that we have provided detailed responses to all questions and concerns raised during the review process. These responses are available both in the updated PDF (which includes a comprehensive rebuttal appendix) and in the comment-by-comment replies within the review system.

We sincerely invite the reviewers to take a look at these materials, as several clarifications and additional analyses directly address the earlier concerns.

If further experiments or explanations would be helpful, please feel free to let us know during the discussion period. We are happy to provide anything that could facilitate your assessment.

Thank you very much for your time and consideration.

Sincerely,

The Authors

---

### Author Response · Authors · 2025-12-03
**Official Comment by Authors (Summary of Concerns and Responses)**

**Dear PCs, SACs, ACs, and Reviewers.**

We sincerely appreciate the time and effort you have devoted to reviewing our paper.
Below, we summarize **(A) our key contributions** and **(B) the main concerns raised during the review process together with our responses**.
For convenience, all updates have been consolidated in Appendix B (titled “**Rebuttal Appendix**”) in the updated manuscript. These revisions will be incorporated into the main paper for the camera-ready version.

Best regards,
The authors


---
**A. Key Contributions:**
1. We present the first integration of event streams into a Diffusion Transformer (DiT)–based model for video frame interpolation (VFI), while prior event-driven VFI methods (e.g., VDM-EVFI) rely on U-Net–based diffusion

2. Based on the contrast maximization principle, we convert events between the two input frames into multiple IWEs and segmented bidirectional optical flows, enabling seamless spatial conditioning and temporal alignment compatible with latent diffusion models. Ablations show clear advantages over alternative event representations.

3. We curate EvPexels, the largest synthetic event–video paired dataset (1,100 pairs), with each video being a single continuous shot and covering diverse motions and scenes.

4. Extensive experiments demonstrate state-of-the-art quantitative and qualitative performance under fair comparisons, highlighting the robustness of our approach.


---

## Reviewer BPER (Rating 4) - Summary

**Concern 1: Unfair comparison with VDM-EVFI**

Response 1: We adapted VDM-EVFI to the Wan 2.1 FLF2V backbone (VDM-EVFI-Wan2.1). Our method clearly outperforms it (Tables 5–6, Figures 11–12 in Appendix B.1).


**Concern 2: Limited qualitative comparisons**

Response 2:   Extensive qualitative results are already included in the original manuscript (Figures 5–9, Appendix A), and the supplementary video further supports them.

**Concern 3: Lack of comparison with non-generative VFI methods**

Response 3:  We added RIFE under consistent evaluation (Tables 5–6 in Appendix B). Our method remains superior.


**Concern 4: Ablation study on Wan 2.1 FLF2V**

Response 4:  The requested ablation study is already included in Table  4 of Appendix A in the original manuscript.


---

## Reviewer 9sZG (Rating 4) - Summary

**Concern 1: Unfair comparison with VDM-EVFI**

Response 1: Same as Reviewer BPER: VDM-EVFI-Wan2.1 is provided, and our method clearly outperforms it (Tables 5–6, Figures 11–12, Appendix B.1).



**Concern 2: Why evaluated on the Davis and Pexels**

Response 2:  We clarified that evaluating on DAVIS and Pexels follows established event-VFI practice and is essential for assessing generalization across diverse motions and scenes beyond what real-event datasets can offer.



**Concern 3: Why the ablation studies are conducted on Pexels and DAVIS rather than on BS_ERGB**

Response 3: Ablations on the BS-ERGB dataset were added in the rebuttal phase (Table 7, Appendix B).

**Concern 4: Ablation studies on layer selection**

Response 4: The layer-selection ablation is included in Table 8, Appendix B.2.

---

## Reviewer uM48 (Rating 6) - Summary

**Concern 1: Additional event‑based conditionings experiments**

Response 1: We evaluated two additional event representations from prior work (Table 9, Appendix B.2). Our IWE & Flow Warping representation consistently outperforms all alternatives across all metrics.

**Concern 2: More description of the EvPexels Dataset**

Response 2: We expanded the dataset description in Appendix B.3.2, including acquisition procedure, scene categories, motion types, and data splits.


---

## Reviewer 6qaC (Rating 4) - Summary

**Concern 1: Discussion on how event stream helps in this framework.**

Response 1: Clarified in Appendix B.3.1, detailing the role and benefits of event streams in our pipeline.

**Concern 2: Novelty of our method.**

Response 2: We explicitly clarified the novelty and key differences from existing event-guided and diffusion-based VFI approaches.


**Concern 3: How we add intermediate frames**

Response 3: Clarified the latent feature update mechanism responsible for generating intermediate frames.

---

### Meta-Review · Area_Chair_etsv · 2026-01-06

**Summary:**

The reviewers’ main concerns focus on fairness and completeness of the evaluation and limited technical novelty. In particular, they note that the method is compared against weaker backbones, making it unclear how much gain comes from the proposed event integration versus the stronger DiT model. They also point to missing or limited comparisons, including with adapted prior methods, non-generative baselines, and alternative event-conditioning strategies, as well as insufficient analysis of the new dataset. Finally, several reviewers feel the approach relies heavily on existing components, with novelty primarily in system integration rather than new theory or algorithms.

**Reviewer Concerns:**

The authors have made a serious effort to address the reviewers’ concerns, for example, by highlighting that some of the suggested experiments were already included in the supplementary material and by adding new experiments and ablation studies. They have also attempted to better clarify the novelty of their work. However, although the proposed approach contains some interesting technical elements and the construction of a new dataset is commendable, the contribution still feels incremental and does not clearly meet the bar for acceptance at this venue.

**Reviewer Scores:**

Reviewer BPER: 4
Reviewer 9sZG: 4
Reviewer uM48: 6
Reviewer 6qaC: 4

The reviewers’ concerns have been partially addressed, particularly those related to comparisons and ablation studies. However, after reviewing the four key contributions highlighted at the end of the rebuttal, I still feel that none of them are especially distinctive. In particular, the novelty of the proposed approach appears to lie mainly in the integration of existing techniques. Therefore, I do not expect the reviewers to significantly increase their scores.

---

### Decision · Program_Chairs · 2026-01-26

Reject